# MutL sliding clamps coordinate exonuclease-independent *Escherichia coli* mismatch repair

Jiaquan Liu[1,4], Ryanggeun Lee[2,4], Brooke M. Britton [1,4], James A. London[1], Keunsang Yang[3], Jeungphill Hanne[1], Jong-Bong Lee[2,3]* & Richard Fishel[1]*

A shared paradigm of mismatch repair (MMR) across biology depicts extensive exonuclease-driven strand-specific excision that begins at a distant single-stranded DNA (ssDNA) break and proceeds back past the mismatched nucleotides. Historical reconstitution studies concluded that *Escherichia coli* (Ec) MMR employed EcMutS, EcMutL, EcMutH, EcUvrD, EcSSB and one of four ssDNA exonucleases to accomplish excision. Recent single-molecule images demonstrated that EcMutS and EcMutL formed cascading sliding clamps on a mismatched DNA that together assisted EcMutH in introducing ssDNA breaks at distant newly replicated GATC sites. Here we visualize the complete strand-specific excision process and find that long-lived EcMutL sliding clamps capture EcUvrD helicase near the ssDNA break, significantly increasing its unwinding processivity. EcSSB modulates the EcMutL–EcUvrD unwinding dynamics, which is rarely accompanied by extensive ssDNA exonuclease digestion. Together these observations are consistent with an exonuclease-independent MMR strand excision mechanism that relies on EcMutL–EcUvrD helicase-driven displacement of ssDNA segments between adjacent EcMutH–GATC incisions.

[1] Department of Cancer Biology and Genetics, The Ohio State University Wexner Medical Center, Columbus, OH 43210, USA. [2] Department of Physics, Pohang University of Science and Technology (POSTECH), Pohang, Gyeongbuk 37673, Korea. [3] School of Interdisciplinary Bioscience and Bioengineering, POSTECH, Pohang, Gyeongbuk 37673, Korea. [4] These authors contributed equally: Jiaquan Liu, Ryanggeun Lee, Brooke M. Britton *email: jblee@postech.ac.kr; rfishel@osu.edu

Mismatch repair (MMR) is a highly conserved excision-resynthesis system that maintains the genome by principally correcting polymerase misincorporation errors[1]. MMR excision commonly begins at a single-stranded DNA (ssDNA) break that marks the error-containing strand and may be several hundred base pairs (bp) distant from the mismatch. Resynthesis of the resulting ssDNA gap is independent of the excision process and is generally completed by the replicative polymerase machinery. Remarkably, in spite of decades of work the cooperative mechanics between multiple independent MMR components that ultimately results in strand-specific excision remain enigmatic. These uncertainties have resulted in a number of competing molecular models for MMR[1–3].

Previous genetic and biochemical analysis established that *E. coli* (Ec) MMR begins with mismatch detection by the prototypical EcMutS protein homodimer[4,5]. Mismatch recognition by EcMutS and its highly conserved MutS homologs (MSH) provoke ATP binding, which triggers the formation of a sliding clamp that randomly diffuses along the mismatched DNA[6–13]. Dissociation from the mismatch permits loading of multiple MutS sliding clamps that ensures redundant lesion identification[1,6–13]. More recent work demonstrated that the EcMutS sliding clamp provides a platform for binding a single N-terminal ATPase domain of similarly conserved EcMutL homologs (MLH/PMS) that in *E. coli* exist as a stable homodimer linked by their C-terminal domains[6,14,15]. Thermal motion of the initial EcMutS–EcMutL complex triggers the remaining unbound EcMutL peptide segments to wrap around the mismatch DNA activating a distinctively different ATP-binding activity that securely links the EcMutL N-terminal domains forming a second sliding clamp[15]. This cascade of extremely stable EcMutS and EcMutL sliding clamps then oscillate between an EcMutS–EcMutL DNA search complex that maintains continuous rotation-coupled contact with the backbone, and separate proteins that display rotation-independent diffusion on DNA and may move independently to a distant site to recreate the EcMutS–EcMutL complex[15–18]. These mechanical progressions enable the two most highly conserved MMR components to robustly explore a mismatched DNA by simple thermal-driven diffusion[19].

The origins of the distant ssDNA break that identifies the error-containing strand responsible for the fidelity of MMR remains poorly understood in most organisms[1]. However, a subset of pathogenic γ-proteobacteria that include *E. coli* recently coopted the DNA adenine methylase (Dam) and MutH to introduce an ssDNA break onto the unmethylated strand of newly replicated transiently hemimethylated GATC sites[20]. Dam/MutH are analogous to other restriction-modification systems[21] where MutH appears similar to the dimeric Sau3A restriction enzyme, except that it functions as a monomer[22,23] and consequently binds poorly to any form of Dam methylated, hemimethylated or unmethylated GATC sites[24]. Single-molecule imaging showed that EcMutH associates with an EcMutL sliding clamp on the mismatched DNA that then recurrently created a rotation-coupled diffusion-mediated EcMutS–EcMutL/EcMutH search complex. The formation of this initial search complex increased EcMutH interactions with the mismatched DNA by at least 1000-fold[15], dramatically enhancing its GATC incision activity[23,25,26].

The evolution of the Dam/MutH system coincides with the conscription of the UvrD helicase as an MMR component[20]. EcUvrD-catalyzed DNA unwinding in vitro is extremely inefficient and must be staged at very low ionic strength to observe activity[27]. Under these conditions EcUvrD performs directional 3'→5' unwinding[27], appears to function as a monomer and/or dimer[28–30], displays very low processivity (~20 bp)[28] and will both unwind and rezip very short duplex DNAs by a strand-switching mechanism with equal frequency[28,31]. While previous single-molecule studies detailed the biophysical events associated with the initiation of *E. coli* MMR, the mechanics of strand-specific excision that begins at the distant MutH-induced ssDNA break remain unknown. Here, we use multiple single-molecule imaging techniques to visualize *E. coli* MMR excision. We find that exceedingly stable ATP-bound EcMutL sliding clamps capture EcUvrD near the ssDNA break, tethering it to the mismatched DNA and dramatically increasing its unwinding processivity. EcSSB regulates the dynamic properties and extent of EcMutL–EcUvrD unwinding. Only a few nucleotides (<50 nt) appear to be removed by ssDNA exonucleases during EcMutL–EcUvrD unwinding events, consistent with a relatively minor role in inhibiting premature DNA ligation of the EcMutH incision before MMR excision can be initiated. These observations suggest that *E. coli* MMR strand-specific excision is unlikely to occur by traditional models that involve extensive exonuclease-mediated excision, but is instead largely performed by EcUvrD helicase-mediated displacement of the mismatch-containing ssDNA strand between adjacent MutH–GATC incisions.

## Results

**EcMutL sliding clamps activate EcUvrD helicase activity**. To examine the strand-specific MMR excision process we constructed an 18.4 kb DNA substrate containing a single G/T mismatch and an ssDNA break located 4.2 kb from the mismatch (Fig. 1a; Supplementary Fig. 1a; Supplementary Table 1). This λ-based DNA substrate is similar to the 6.4 kb bacteriophage f1 DNA used in the original MMR reconstitution studies that also contained a G/T mismatch and a Dam/EcMutH GATC recognition/incision site located 1 kb distant from the mismatch[32,33]. The longer distance between the mismatch and the ssDNA break in our mismatched DNA substrate insures that single protein molecules can be clearly resolved between these two sites. Single 18.4 kb DNA molecules were stretched across a passivated flow cell surface by controlled laminar flow and linked at both ends via biotin-neutravidin (Supplementary Fig. 1b, c). Prism-based single molecule total internal reflection fluorescence (smTIRF) microscopy combined with tracking and analysis of purified fluorophore-labeled EcMutS, EcMutL, and EcMutH has been previously described by our group[15,34]. For the MMR excision reaction EcUvrD, EcSSB, EcExoI, and EcRecJ were also purified (Supplementary Fig. 1d; Methods). Genetically *wild type* FGE-tagged EcUvrD (EcUvrD-his$_6$-ald$_6$; Supplementary Table 2) was labeled with AlexaFluor (AF) 647 utilizing hydrazinyl-iso-pictet-spengler (HIPS) chemistry[34], while EcSSB containing a single Cys residue was labeled with Cy3 using maleimide chemistry (Supplementary Fig. 1d). Protein labeling with small chemical fluorophores permits the crucial formation of mismatch- and EcMutS-dependent ATP-bound EcMutL sliding clamps that diffuse rapidly along the entire length of the 18.4 kb mismatched DNA (Fig. 1b)[15]. These extremely stable MLH/PMS sliding clamps appear to be absent when other antibody-based or larger fluorophore-labeling schemes are utilized[35–38].

We observed significant co-localization of EcMutL sliding clamps with EcUvrD on the mismatched DNA that resulted in visibly coordinated directional motion (Fig. 1c, Supplementary Fig. 2a, b). The majority of initial EcMutL–EcUvrD co-localization events were within 200-400 nm of the defined strand break (Supplementary Fig. 2c), which appears close to the spatial resolution of this imaging system. These results are consistent with the conclusion that EcMutL interacts with EcUvrD at or very near the strand break. Co-localization of EcMutL with EcUvrD required mismatch recognition by EcMutS[4,5] and displayed a similar frequency regardless of whether the ssDNA break was

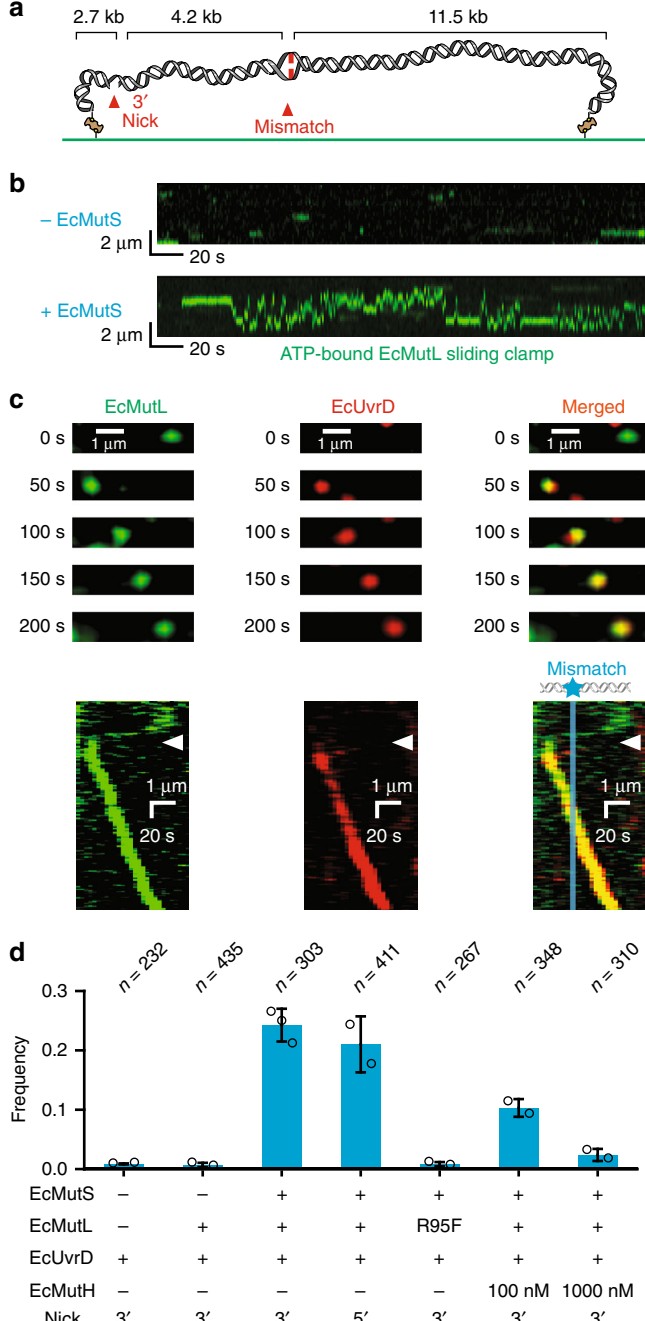

**Fig. 1** The EcMutL sliding clamp activates processive EcUvrD helicase Unwinding. **a** An illustration of the 18.4-kb mismatched DNA with a 3' nick in the single molecule total internal reflection fluorescence (smTIRF) system. **b** Representative kymographs showing the absence of any AF555-EcMutL sliding clamps without EcMutS (−EcMutS, top) and the formation of an extremely stable[15] ATP-bound AF555-EcMutL sliding clamp on the mismatched DNA in the presence of EcMutS (10 nM, +EcMutS, bottom). **c** Representative fluorescent images (top) and kymographs (bottom) showing an EcMutL–EcUvrD complex unwinds the mismatched DNA in the presence of EcMutS (10 nM) and ATP (1 mM). AF555-EcMutL is shown in green and AF647–EcUvrD is shown in red. Arrowheads indicate association of EcMutL sliding clamp with EcUvrD. Blue line and star indicate the mismatch position. **d** The frequency of EcUvrD unwinding under various conditions observed by smTIRF (mean ± s.d.; n = number of DNA molecules; Methods). Open circles represent the frequencies from separate experiments at the indicated conditions.

located 3' or 5' of the mismatch (Fig. 1d)[39]. No co-localization or directional motion occurred in the absence of EcMutL or when the ATP binding-deficient EcMutL(R95F) was substituted for *wild type* EcMutL, suggesting that the prerequisite formation of an ATP-bound EcMutL sliding clamp is essential for co-localization (Fig. 1d)[15]. Both the fast-diffusing EcMutL sliding clamp (Supplementary Fig. 2a)[15] and the slower-diffusing EcMutS–EcMutL complex (Supplementary Fig. 2b)[15] co-localized and activated EcUvrD movement.

The EcMutL–EcUvrD progression along the DNA often exceeded the 4.2 kb distance from the ssDNA break to the mismatch and frequently terminated in a double-strand break (DSB; Supplementary Fig. 2a, b). The formation of DSBs supports the notion that co-localization is accompanied by extensive EcUvrD catalyzed unwinding activity that sporadically encounters a random ssDNA break on the opposite strand. These random ssDNA breaks appear to arise from construction/handling of the 18.4 kb mismatch DNA substrate, which also produce a small background of unwinding events that do not start near the defined ssDNA break (Supplementary Fig. 2c). We observed an approximately equivalent frequency of unwinding toward the mismatch compared to away from the mismatch, suggesting that the interaction between EcMutL and EcUvrD may occur on either the 3' or 5' side of the strand break (Supplementary Fig. 2d). We noted that the intensity of EcUvrD fluorescence nearly always increased with increasing tract length (Fig. 1c; Supplementary Fig. 2a, b). This observation strongly suggest that multiple EcUvrD molecules may become associated with the ssDNA during the unwinding process.

Increasing the EcMutH concentration appeared to inhibit the frequency of EcUvrD unwinding events (Fig. 1d). However, competition was only observed when EcMutH substantially exceeded the concentration of EcUvrD, which does not occur in vivo where EcUvrD is at least 50-fold in excess of EcMutH[40–42]. Unlike the stable freely diffusing EcMutL–EcMutH complex observed in previous studies[15], we only observed an EcMutL–EcUvrD complex on the mismatched DNA when it is unwinding the DNA. Taken together, these results are consistent with the hypothesis that EcMutH and EcUvrD share overlapping interaction sites with EcMutL, with an EcMutL–EcUvrD–ssDNA unwinding interface that is generally preferred because of the overwhelming EcUvrD cellular concentration.

**EcMutL tethers EcUvrD to the DNA increasing its processivity**. To quantitatively examine MMR unwinding kinetics we turned to single molecule flow-stretching (smFS) analysis[43]. In this system, one end of the mismatched DNA is bound to the flow-cell surface by biotin-streptavidin while the other DNA end is linked to an anti-digoxygenin antibody-coated 2.8 μm super-paramagnetic bead (SPM) via a 5'-digoxigenin (Fig. 2a, top; Supplementary Table 1; Methods). The combination of laminar flow ($F_{flow}$) and magnetic ($F_{mag}$) forces on the SPM bead results in a net stretching force on the mismatched DNA (Fig. 2a, top)[43]. At low force (2.5 pN) a duplex DNA is near fully extended because of its worm-like chain properties[44]. Helicase unwinding or exonuclease digestion results in the production of ssDNA, which assumes freely jointed chain characteristics with dramatically reduced the DNA extension under similar low-force conditions (Fig. 2a, bottom, red and green arrows)[45]. We determined that the smFS system can resolve 20 nm movements at 1 s time resolution, which translates to the formation of ~60 nt of naked ssDNA or ~80 nt ssDNA bound by EcSSB (Methods)[43].

Including EcMutS, EcMutL, and EcUvrD resulted in numerous SPM beads that displayed punctuated movement indicating the formation of ssDNA (Fig. 2b). The production of ssDNA required

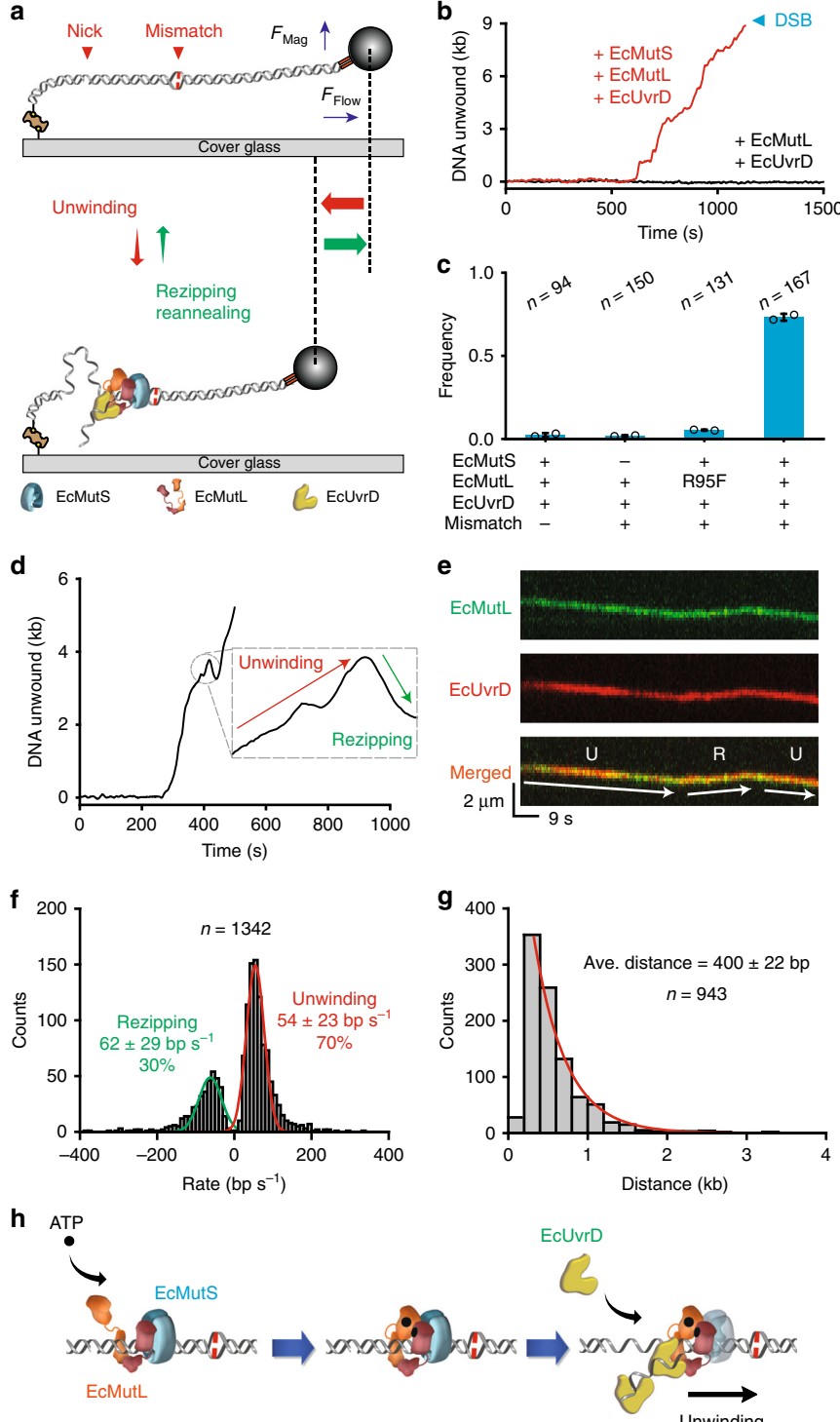

a mismatch, EcMutS, ATP-binding by EcMutL (Fig. 2c, see - and R95F, respectively; Supplementary Fig. 3a) and occasionally terminated in a DSB (Fig. 2b). These observations qualitatively match the smTIRF studies (Fig. 1) and support the conclusion that the cascade loading of ATP-bound EcMutS and EcMutL sliding clamps is essential for the processive unwinding activity exhibited by the EcUvrD helicase[15]. Similar mismatch-, EcMutS- and EcMutL-dependent activation of EcUvrD helicase activity has been observed under reduced ionic strength conditions in bulk biochemical studies[46]. However, we found that with EcMutS and EcMutL, the EcUvrD DNA unwinding activity was robust at

physiological ionic strength and displayed significantly longer lifetimes than studies with EcUvrD alone[28,31]. Embedded in the punctuated SPM bead movement were tracts that represented unwinding and rezipping which appeared similar to previous EcUvrD studies (Fig. 2d)[28,31]. The rezipping of unwound DNA likely results from strand-switching by EcUvrD that maintains its intrinsic 3'→5' directional activity[28,31]. Comparable unwinding-rezipping events containing co-localized EcMutL and EcUvrD could also be detected by smTIRF (Fig. 2e). However, unlike prior studies[28,31] we found that the frequency of EcUvrD unwinding was greater than rezipping in the presence of the EcMutL sliding

**Fig. 2** The EcMutL–EcUvrD complex unwinds and rezips mismatched DNA. **a** A schematic illustration of the DNA substrate shown in Fig. 1 tethered to the surface at one end (biotin-streptavidin) and containing a super paramagnetic bead bound to the other end (digoxigenin-antidigoxigenin) in a single molecule flow-stretching (smFS) system. DNA unwinding by EcUvrD results in single-stranded regions that shorten the mismatched DNA, while rezipping restores the DNA length. **b** Representative time trajectories of SPM beads tethered to smFS mismatched DNA during unwinding (red) in the presence of EcMutS (100 nM), EcMutL (100 nM) and EcUvrD (20 nM); black, in the presence of EcMutL (100 nM) and EcUvrD (20 nM) only. Blue arrowhead indicates when DNA was broken forming a double-strand break (DSB). **c** The frequency of EcUvrD unwinding in the presence of various protein components by smFS (mean ± s.d.; $n$ = number of DNA molecules; Methods). Open circles represent the frequencies from separate experiments at the indicated conditions. **d** Representative time trajectory of the SPM beads illustrating unwinding and rezipping events on a mismatched DNA in the presence of EcMutS (100 nM), EcMutL (100 nM) and EcUvrD (20 nM). A magnification of the rezipping trajectory is shown in the inset. **e** Representative fluorescent kymographs illustrating unwinding (U) and rezipping (R) events on a mismatched DNA in the presence of EcMutS (10 nM, unlabeled), AF555-EcMutL (20 nM, green) and AF647–EcUvrD (20 nM, red). **f** Histogram of binned unwinding/rezipping rates that were fit to Gaussian function to derive the average rates (mean ± s.d.; $n$ = number of events). **g** Histogram of binned punctuated unwinding distance that were fit to a single exponential decay to derive the average distance (mean ± s.e.; $n$ = number of events; Supplementary Fig. 3b). **h** Illustration of ATP-bound EcMutL sliding clamp (brown) loaded by an EcMutS sliding clamp (blue) that tethers EcUvrD (yellow) near the ssDNA break either as an EcMutL sliding clamp alone or as an EcMutS–EcMutL complex to promote extensive DNA unwinding.

clamp (Fig. 2f). The average distance between punctuated unwinding events (400 ± 22 nt; Fig. 2g; Supplementary Fig. 3b) and single unwinding events collected at dramatically reduced EcUvrD concentration (424 ± 73 nt; Supplementary Fig. 3c, d) were largely identical. These results suggest that EcMutL–EcUvrD unwinding processivity is at least 10-fold greater than EcUvrD alone[28]. In contrast, the kinetics of unwinding and rezipping (54 ± 23 bp s$^{-1}$ and 62 ± 29 bp s$^{-1}$, respectively; Fig. 2f) appeared comparable to previous studies[28,31]. Taken as a whole these observations are consistent with the conclusion that the EcMutL sliding clamp does not alter the catalytic properties of EcUvrD, but rather enhances helicase processivity by tethering EcUvrD to the mismatched DNA[47] that in-turn appears to provide an asymmetric configuration that favors unwinding over rezipping (Fig. 2h).

**EcSSB restrains EcMutL–EcUvrD strand displacement**. The addition of EcSSB dramatically altered the progressions of the MMR components on the mismatched DNA (Fig. 3). When visualized by smTIRF numerous events were detected where an EcMutL sliding clamp co-localized with EcUvrD near the ssDNA break. However, these complexes displayed very little if any detectable movement (compare Fig. 1c with Fig. 3a and Supplementary Fig. 2e). In contrast, numerous unwinding and rezipping events were detected on single molecules by smFS (Fig. 3b). The frequency of DNA molecules undergoing unwinding/rezipping was relatively constant over a wide-range of EcSSB concentrations (Fig. 3c, top). We parsed the molecules observed at each EcSSB concentration into those that had undergone unwinding followed by rezipping back to fully duplex DNA (restored) and those that remained unwound after the 25 min observation period (unwound; Fig. 3c, bottom). The majority of mismatched DNA molecules at zero or sub-saturating concentrations of EcSSB (<10 nM) displayed very long ssDNA tracts, remained unwound, and occasionally terminated in a DSB similar to the smTIRF studies in the absence of EcSSB (Fig. 3c). However, at saturating EcSSB concentrations (>200 nM) the majority of the mismatched DNA molecules were rezipped to restore their original length (Fig. 3b, c).

To visualize the real-time formation of ssDNA we examined the binding of Cy3-labeled EcSSB to mismatched DNA undergoing EcMutL–EcUvrD unwinding/rezipping by smTIRF. Two concentration of Cy3-EcSSB (10 nM and 200 nM) were analyzed that seemed to generated molecules that displayed either a majority unwound or restored events by smFS (Fig. 3c). Extended Cy3-EcSSB binding tracts were detected on the mismatched DNA at sub-saturating EcSSB (10 nM; Fig. 3d, e, left). These observations appeared comparable to the long unwinding tract seen in the absence of EcSSB (compare Fig. 1c to Fig. 3d, e, left), and suggest that multiple EcUvrD binding events and extensive EcUvrD catalyzed unwinding may occur when regions of ssDNA persist. Under these conditions the spatial resolution of the smTIRF system makes Cy3-EcSSB fluorescence appear relatively continuous along the unwound tracts. In contrast, at saturating EcSSB (200 nM) the Cy3-EcSSB binding tracts appeared as foci that increased and decreased in intensity (Fig. 3d, e, right). These results are consistent with the real-time gain and loss of bound Cy3-EcSSB, respectively, and suggest that EcSSB binding and dissociation to ssDNA is coincident with the restricted EcMutL–EcUvrD unwinding and rezipping events observed by smFS (Figs. 2b and 3b).

While unwinding was still favored in the presence of EcSSB, the frequency of rezipping events markedly increased (compare Fig. 4a to Fig. 2f); principally accounting for the increased restoration of the mismatched DNA to its original length (Fig. 3c). The processivity in the presence of EcSSB (382 ± 14 bp; Fig. 4b) appeared similar to the absence of EcSSB (compare Fig. 4b to Fig. 2g), while the rate of unwinding and rezipping (51 ± 21 nt s$^{-1}$ and 74 ± 41 nt s$^{-1}$, respectively; Fig. 4a) was consistent with both the absence of EcSSB (compare Fig. 4a to Fig. 2f) and previous studies with EcUvrD alone[28,31]. The relative invariance of unwinding-rezipping rates suggests that the fundamental kinetics of the tethered EcUvrD helicase activity remain unchanged over a variety of biochemical conditions. Importantly, EcUvrD-catalyzed DNA rezipping activity seems capable of easily displacing EcSSB bound to ssDNA, an activity that appears similar to the related eukaryotic helicase RAD54 that dislodges RAD51 filaments from DNA[48].

The maximum unwinding distance (MUD) at multiple EcUvrD concentrations and in the presence of EcSSB was extracted by examining the cycles of unwinding-rezipping events on single mismatched DNA molecules over the entire observation period (25 min; Fig. 4c). We found that the MUD appeared to saturate (MUD$_{sat}$) at an EcUvrD concentration (20–100 nM) that was ~30-fold less than the cellular concentration (1657 ± 243 bp; Fig. 4d; Supplementary Fig. 4a–d). This distance is significantly shorter than in the absence of EcSSB (Fig. 2b, d) and explains the general lack of observable EcMutL–EcUvrD motion by smTIRF (Fig. 3a, d, e, Supplementary Fig. 2e), since the MUD$_{sat}$ overlaps the spatial resolution of this imaging method. To test the hypothesis that multiple EcUvrD proteins might load at exposed ssDNA regions in the absence of EcSSB, we examined unwinding at dramatically reduced EcUvrD concentration (1 nM) in the absence of EcSSB and found the MUD (505 ± 76 bp; Fig. 4d, green; Supplementary Fig. 4e) was significantly shorter than at

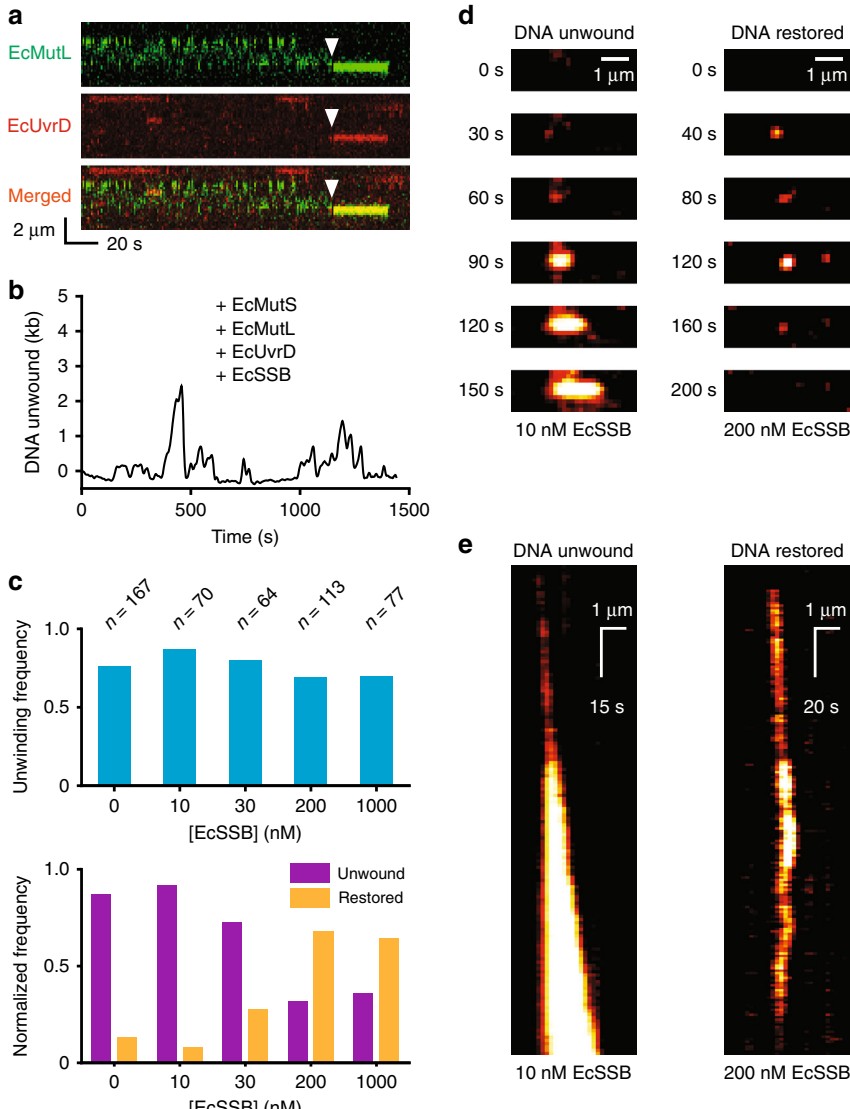

**Fig. 3** EcSSB modulates EcUvrD unwinding. **a** Representative kymographs showing the formation of a AF555-EcMutL(green)-AF647–EcUvrD(Red) complex (merged, yellow) in the presence of ATP (1 mM), EcMutS (10 nM, unlabeled) and EcSSB (200 nM, unlabeled). Arrowheads indicate the association of an EcMutL sliding clamp with EcUvrD. **b** Representative smFS time trajectory of an SPM beads showing multiple DNA unwinding and rezipping events in the presence of EcMutS (100 nM), EcMutL (100 nM), EcUvrD (20 nM), and EcSSB (200 nM). **c** The frequency of DNA unwinding (top), normalized frequency of unwound DNA (Fig. 2b, d; Methods) and restored DNA (**b**; Methods) observed by smFS in the presence of EcMutS (100 nM), EcMutL (100 nM), EcUvrD (20 nM) and different concentrations of EcSSB (bottom; $n$ = number of DNA molecules). **d, e** Representative individual fluorescent images and complete kymographs, respectively, showing the formation of Cy3-EcSSB coated ssDNA during unwinding. Data were acquired by smTIRF in the presence of EcMutS (100 nM), EcMutL (100 nM), EcUvrD (10 nM) and subsaturating (left; DNA Unwound, 10 nM) or saturating (right; DNA Restored, 200 nM) concentrations of EcSSB (see Methods).

elevated EcUvrD. Taken together, we conclude that EcSSB moderates extensive unwinding by excluding secondary EcUvrD binding and/or multimerization on previously displaced ssDNA regions (Fig. 4e).

**Exonuclease digestion rarely accompanies EcUvrD unwinding**. A permanent shortening of the SPM bead position following EcMutL–EcUvrD unwinding-rezipping is a hallmark of ssDNA exonuclease-dependent excision (Fig. 5a, red-dashed lines; Supplementary Fig. 5a). Surprisingly, extensive excision of the unwound ssDNA was rare when a single or combination of three canonical MMR ssDNA exonucleases were included in the MMR reactions (Fig. 5b, Supplementary Fig. 5b; Supplementary Table 3). These results differ substantially from bulk exonuclease

activity analysis that shows EcExoI rapidly degrades a purified ~2 knt ssDNA ($t_{0.5}$ = 1.3 min), which occurred significantly faster in the presence of EcSSB ($t_{0.5}$ < 0.4 min)[49] regardless of incubation temperature or the presence of EcUvrD (Supplementary Fig. 6a, b). EcRecJ and EcExoVII also actively degrade the ~2 knt ssDNA ($t_{0.5}$ = 4.7 min and $t_{0.5}$ = 1.1 min, respectively), although slightly slower than EcExoI (Supplementary Fig. 6c). Taken together we conclude that the lack of extensive excision in the presence of ensemble MMR components is not related to underlying biochemical conditions, the type of ssDNA exonuclease or the addition of EcSSB and EcUvrD.

It is possible that the inclusion of ssDNA exonucleases might alter the ability to detect the formation of ssDNA by smFS. To eliminate this potential issue we examined the formation of Cy3-EcSSB bound ssDNA tracts by smTIRF with all the consensus

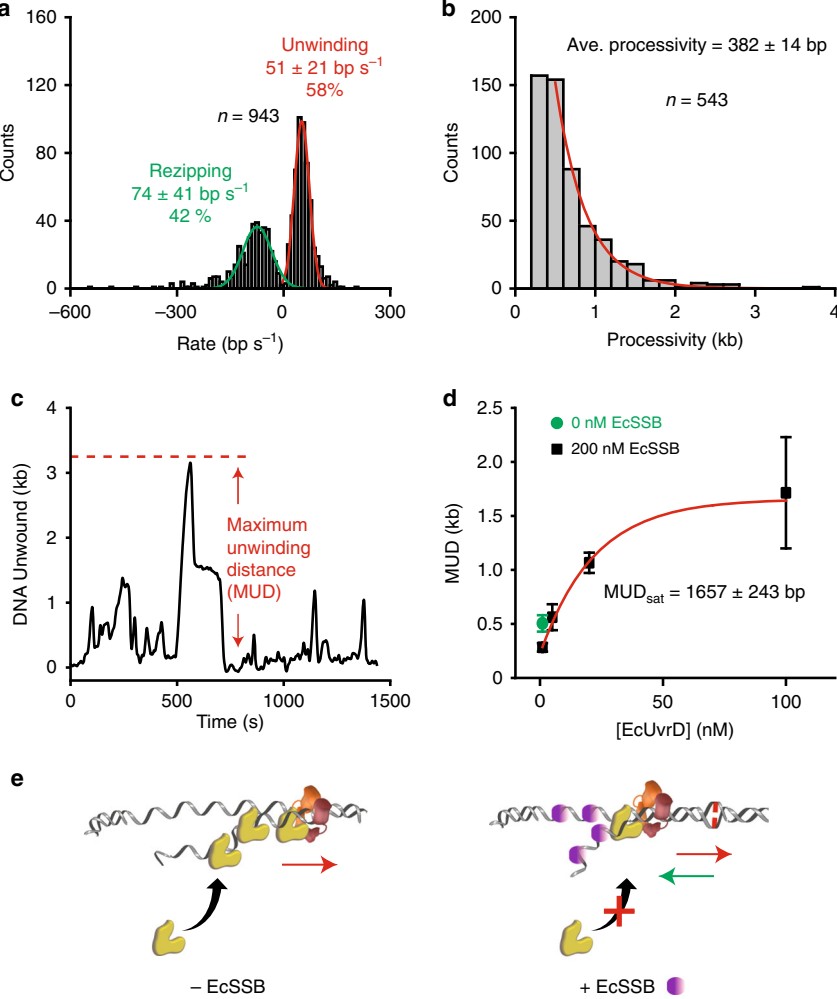

**Fig. 4** EcSSB prevents additional EcUvrD binding to the displaced ssDNA. **a** Histogram of binned unwinding and rezipping rates in the presence of EcMutS (100 nM), EcMutL (100 nM), EcUvrD (20 nM), and EcSSB (200 nM) that were fit to Gaussian functions to derive the average rates (mean ± s.d.; $n$ = number of events). **b** Histogram of binned unwinding processivity in the presence of EcMutS (100 nM), EcMutL (100 nM), EcUvrD (20 nM), and EcSSB (200 nM) that were fit to a single exponential decay to derive the average processivity (mean ± s.e.; $n$ = number of events). **c** An illustration of the maximum unwinding distance (MUD) during a single molecule cycle over 25 min. **d** Maximum unwinding distance (MUD) measured under various conditions (mean ± s.e.; Supplementary Fig. 4). Data were fit by single exponential equation to derive the MUD at saturated EcUvrD concentration (MUD$_{sat}$, red line; mean ± s.e.). **e** An illustration of EcUvrD loading onto a mismatched DNA in the absence of EcSSB (left) compared to in the presence of EcSSB (right). EcSSB is shown in purple.

MMR components present (Fig. 5c; Supplementary Fig. 6d). Non-excision on the mismatched DNA appears as Cy3-EcSSB foci that increase and decrease in intensity similar to Cy3-EcSSB foci in the absence of the ssDNA exonucleases (compare Fig. 5c, left with Fig. 3d, e, right). In contrast, stable excision tracts in the mismatched DNA retain the Cy3-EcSSB foci over the entire observation period (Fig. 5c, right). We found that the fraction of marked Cy3-EcSSB excision tracts did not significantly change in the presence or absence of the ssDNA exonucleases (9% versus 13%; Fig. 5d). Combined with the smFS and bulk biochemical studies, these results are consistent with the conclusion that the cooperative EcMutS, EcMutL, and EcUvrD activities on mismatch DNA are not accompanied by extensive ssDNA exonuclease digestion.

While extensive excision was not observed, we detected what appeared to be very short excision events associated with some cycles of EcUvrD unwinding-rezipping (Fig. 5a, red-dashed lines). To determine whether these events were significant, we first evaluated the change in average SPM bead positions (baseline drift) following numerous rezipping events at multiple EcUvrD

concentrations in the absence of ssDNA exonuclease (47 ± 5 nt; Fig. 5a; Supplementary Fig. 5b; Methods). Including EcExoI resulted in a slight increase in the apparent excision at all EcUvrD concentrations (Fig. 6a; Supplementary Fig. 5b; Supplementary Table 3) that is most evident in the outlier events displaying longer tracts than the mean excision length (Fig. 6a). After subtracting the baseline drift in the absence of ssDNA exonucleases, we found a consistent but very small increase in excision length that decreased to ~45 nt with increasing EcUvrD (Fig. 6b; Supplementary Fig. 5b; Supplementary Table 3).

We examined the possibility that the excision distance might be related to the time the unwound strand exists in the ssDNA state (Fig. 6c). Remarkably, increasing EcUvrD resulted in ssDNA exposure dwell times that were modestly longer, while the excision lengths collapsed to nearly the background density distribution in the absence of EcExoI (Fig. 6d). This concentration-dependent pattern of reduced excision appears to suggest that EcUvrD may at least partially shield the displaced ssDNA end from exonuclease digestion similar to the related EcRep helicase[50]. A comparable pattern of very short excision

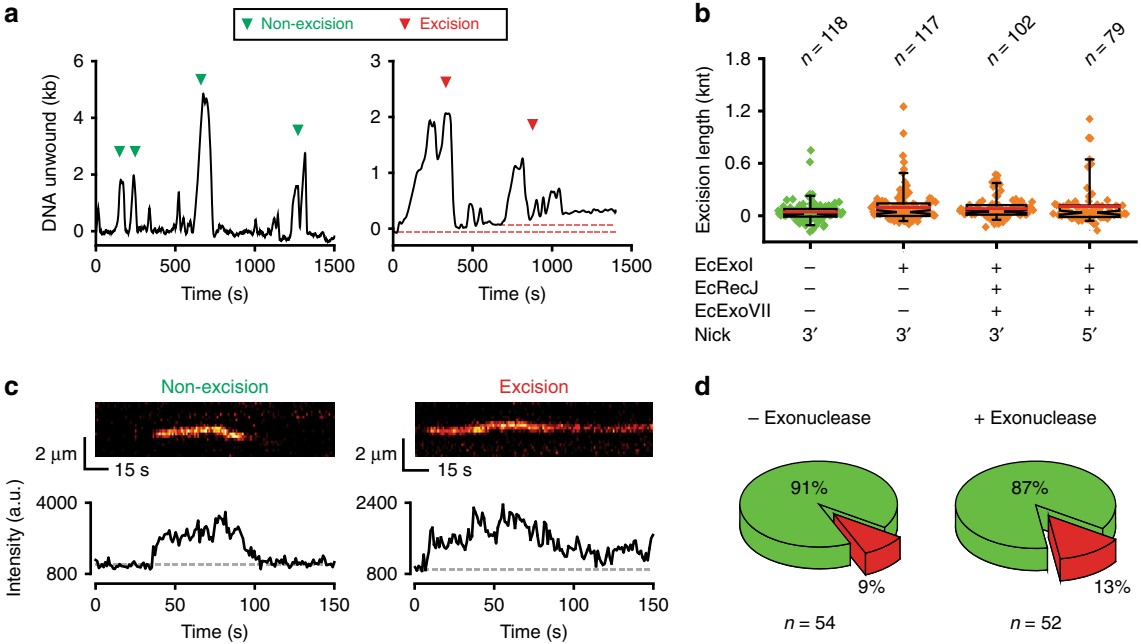

**Fig. 5** Extensive exonuclease excision is rare during ensemble MMR. **a** Representative smFS time trajectories of the SPM beads in the presence of EcMutS (100 nM), EcMutL (100 nM), EcUvrD (20 nM), EcSSB (200 nM), and EcExoI (20 nM; Methods). The unwinding followed by rezipping are indicated either as non-excision events (green arrowheads) or excision events (red arrowheads). The baseline of each excision event is indicated by a red dotted line. **b** Box plots of excision lengths in the presence of different ssDNA exonucleases showing the mean (red line), median (indentation), the upper and the lower quartiles (box ends) and the outliers (5% and 95% whiskers). Diamonds indicate individual events (n = number of events). **c** Representative kymographs (top) and time-dependent fluorescent intensities (bottom) of Cy3-EcSSB during ensemble MMR. Data were acquired by smTIRF in the presence of EcMutS (100 nM), EcMutL (100 nM), EcUvrD (20 nM), Cy3-EcSSB (200 nM) and three ssDNA exonucleases (Methods). Gray dots indicate the baseline fluorescent intensity. The Cy3-EcSSB binding followed by complete dissociation indicates the non-excision (left); the Cy3-EcSSB binding followed by partial dissociation indicates the excision (right). **d** Pie charts showing the distributions of the non-excision events (green) and excision events (red) in the absence (left) or presence (right) of the three ssDNA exonucleases.

tracts was observed when three canonical ssDNA exonucleases were included in the complete MMR reaction (Supplementary Fig. 5b; Supplementary Table 3). Together these results underscore the conclusion that the extensive exonuclease digestion envisioned for *E. coli* strand-specific MMR excision by prior reconstitution studies[32] is exceedingly rare.

**MMR excision is performed by EcUvrD strand displacement**. If extensive exonuclease digestion is rare, how is MMR accomplished? We surveyed the *E. coli* genome and determined the average distance between adjacent Dam/MutH GATC recognition sites (227 ± 3 bp; Fig. 7a). Remarkably, 99.8% of these adjacent GATC segment lengths (GATC→GATC) are shorter than the $MUD_{sat}$ of ensemble MMR component unwinding events (Fig. 7b; Fig. 4d; Supplementary Fig. 4a–d). Previous single-molecule imaging studies indicated that the EcMutS–EcMutL–EcMutH search complex could interrogate ~12 kb of naked DNA[15]. Combined with redundant EcMutS and EcMutL complexes loaded onto a mismatched DNA[1,6], it seems likely that EcMutH will introduce a strand break at nearly every hemimethylated GATC site in a large region surrounding the mismatch. This idea is supported by recent biochemical studies that demonstrated adjacent GATC sites are rapidly incised[51] and multiple GATC sites enhance MMR[26].

The presence of numerous GATC strand breaks on the mismatched DNA suggests that a tethered EcMutL–EcUvrD complex could unwind and displace an error-containing strand between adjacent GATC sites. To test the hypothesis, we constructed a DNA substrate containing ssDNA strand breaks flanking the mismatch (Fig. 7c; Methods). We observed efficient ssDNA excision (Fig. 7d), with the majority of tracts exhibiting a

length comparable to the distance between the dual strand breaks (Fig. 7e; 901 ± 197 nt compared to a calculated 927 nt). Taken together, these results strongly suggest that EcMutL-tethered EcUvrD helicase-driven strand displacement can complete the vast majority of *E. coli* MMR excision events in the absence of an ssDNA exonuclease.

## Discussion

The ensemble single-molecule imaging results presented here have added significant details as well as surprising mechanics to MMR strand-specific excision that is activated by the cascade of EcMutS and EcMutL sliding clamps. For example, previous studies had detected a physical interaction between EcMutL with EcMutH[25] and EcMutL with EcUvrD[46,52]. Here we have uncovered a binding competition between EcMutH and EcUvrD with the EcMutL sliding clamp suggesting overlapping interaction domains. In theory, the overwhelming cellular concentration of EcUvrD should favor an interaction with EcMutL sliding clamps. However, an EcMutL–EcUvrD unwinding complex cannot be assembled without a prior strand break introduced by the EcMutS/EcMutL–EcMutH search complex[15]. Only when a strand break is present can EcUvrD initiate short repetitive unwinding-rezipping events[28,31] that may ultimately be captured by a long-lived EcMutL sliding clamp. These essential biophysical progressions appear to ensure a functional ordering of interactions that still displays dynamic and stochastic mechanics. Moreover, overlapping interaction domains might hypothetically explain the evolution of the Dam/MutH MMR system, which appeared to near simultaneously mutate an essential endogenous MutL-endonuclease and conscript both MutH and UvrD, which appear to be the functional equivalents of the MutL-endonuclease[20]. A

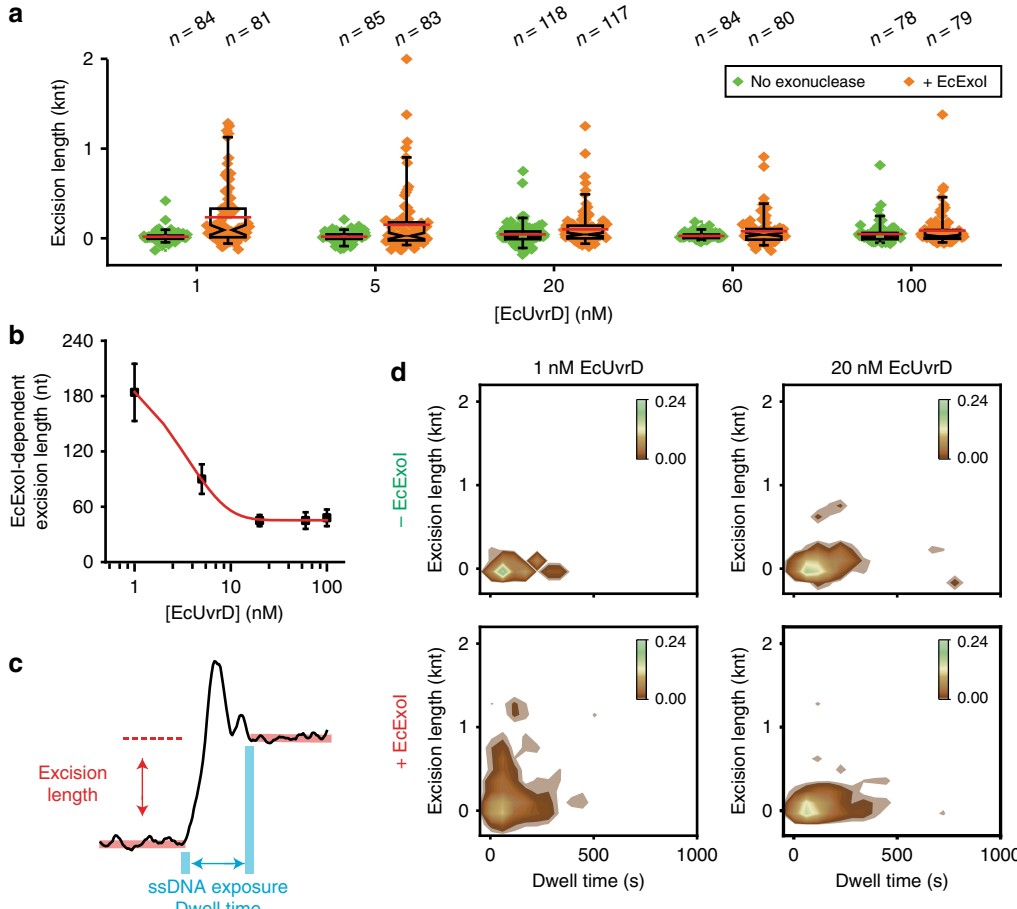

**Fig. 6** EcUvrD partially shields the displaced ssDNA end from exonuclease digestion. **a** Box plots of excision lengths at various EcUvrD concentrations in the absence or presence of EcExoI (20 nM). Mean (red line), median (indentation), the upper and the lower quartiles (box ends) and the outliers (5% and 95% whiskers) are indicated. Diamonds indicate individual events ($n$ = number of events). **b** A plot of EcExoI-dependent excision length at various concentrations of EcUvrD following exponential curve fitting (mean ± s.e.; Methods; Supplementary Fig. 5b). **c** An illustration of the ssDNA exposure dwell time and excision length determination during an unwinding-rezipping cycle. **d** 2D kernel density plots of excision length (vertical) versus ssDNA exposure dwell time (horizontal) in the presence of EcUvrD (1 nM, left) and EcUvrD (20 nM, right) without (top) and with (bottom) EcExoI (20 nM).

detailed identification of the interaction domains between EcMutL with EcMutH and EcUvrD will be required to confirm this hypothesis.

We found that EcSSB moderates extensive unwinding by excluding secondary EcUvrD helicase binding onto the exposed ssDNA. This unwinding control may help to limit possible long-range EcMutL–EcUvrD unwinding encounters with spurious single strand breaks that might occur within the cell and contribute to detrimental DSBs. The presence of EcSSB also led to an altered global EcUvrD unwinding-rezipping activity such that it nearly always restored the mismatched DNA to its original length. Previous single-molecule studies would have been unable to detect similar EcSSB regulation since the unwinding processivity of EcUvrD alone is ~20 nt[28,31], which is smaller than the binding site size of EcSSB[53].

The lack of extensive ssDNA exonuclease digestion during ensemble MMR was unexpected. These results dramatically limit traditional exonuclease-dependent *E. coli* MMR models, which appears antithetical to decades of doctrine[3,54]. It is formally possible that another *E. coli* exonuclease(s) might perform ssDNA excision during MMR in spite of abundant genetic and biochemical discoveries over the years. For the single-molecule imaging studies reported here, we included three of the four consensus MMR ssDNA exonucleases in the complete strand-specific excision reactions. The fourth ssDNA exonuclease,

EcExoX, is ~10-fold less active than EcExoI and perhaps the least potent ssDNA exonuclease in *E. coli*[55]. Thus, we consider it unlikely that additional and/or redundant ssDNA exonucleases might catalyze the extensive ssDNA excision envisioned by conventional MMR models.

Curiously, the original *E.coli* reconstitution studies linked the ssDNA exonuclease EcExoI to MMR as a factor that competed with DNA ligase[32]. In the absence of EcExoI, DNA ligase appeared to seal the EcMutH-induced GATC strand-breaks faster than MMR excision could be accomplished in vitro[32]. The uncoupled very short ssDNA excision tracts observed here support the idea that the principal role for ssDNA exonucleases is to inhibit premature DNA ligation until MMR is completed. A relatively minor role in MMR appears to clarify puzzling historical genetic data that showed simultaneous mutation of the canonical *E. coli* ssDNA exonucleases EcExoI, EcExoVII, EcRecJ, and EcExoX merely increased spontaneous mutation rates by 7-fold, while mutation of any core MMR component (EcMutS, EcMutL, EcMutH, EcUvrD) increased spontaneous mutation rates by at least 100-fold (Supplementary Table 2)[56]. We also note that excision catalyzed the ssDNA exonucleases starts by releasing nucleotides from the EcMutH GATC incision site which has the added effect of inhibiting the ability of Dam to recognize and methylate the newly replicated strand until MMR is completed.

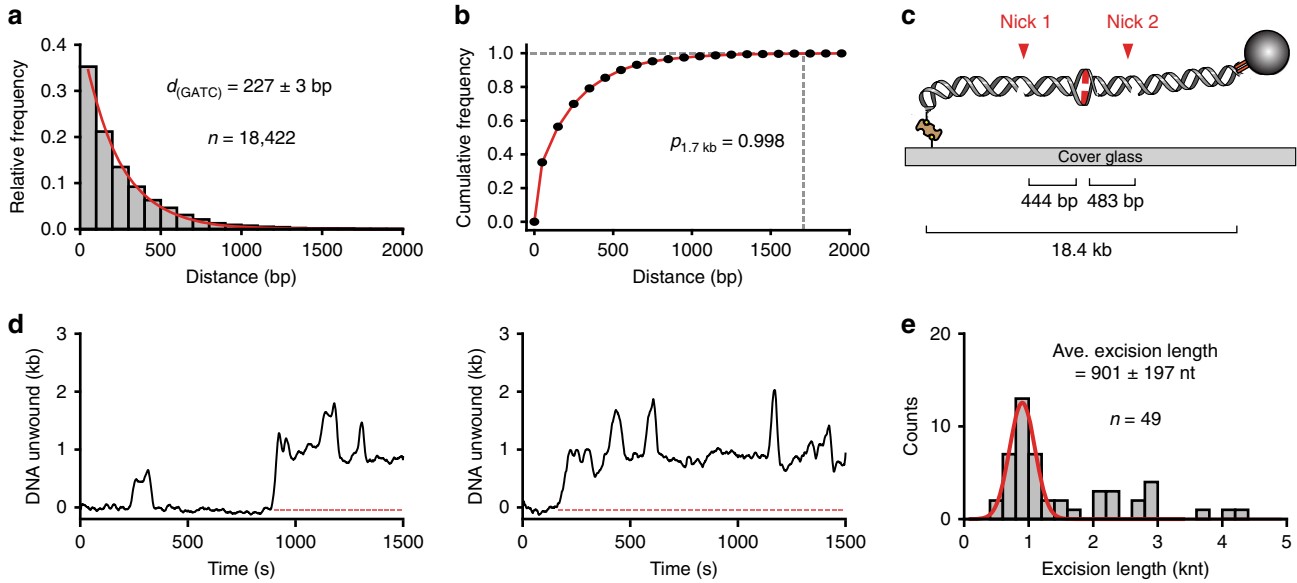

**Fig. 7** EcUvrD removes the mismatch-containing strand between two GATC sites. **a** Histogram of binned distance between adjacent GATC sites in *E. coli*. **b** Cumulative frequency between adjacent GATC sites in *E. coli*. Dotted line indicates that 99.8% of the adjacent GATC sites have a distance that is less than MUD$_{sat}$ (Fig. 4d). **c** An illustration of the smFS DNA substrate containing a mismatch between two adjacent strand breaks. **d** Representative time trajectories of the SPM beads showing the EcUvrD-mediated strand excision between two strand breaks. smFS experiments were performed with a DNA substrate (**c**) in the presence of EcMutS (100 nM), EcMutL (100 nM), EcUvrD (20 nM), EcSSB (200 nM) and the absence of ssDNA exonucleases. A baseline increase following rezipping indicates an excision event (red dotted line: baseline of an excision event). **e** Histogram of binned excision length in (**d**) that were fit to a Gaussian function (red line; mean ± s.d.; *n* = number of events).

The modest mutator effect in vivo contrasts the complete MMR defect when extracts deficient in the four conventional ssDNA exonuclease were examined in vitro[56]. The original reconstitution studies, as well as virtually all subsequent biochemical studies of *E. coli* MMR, utilized DNA substrates containing a single DNA mismatch and a single distant (1 kb) GATC site[32]. With this DNA substrate MMR exhibited exonuclease-dependent excision that began at the EcMutH-GATC strand break and terminated randomly just past the mismatch[32,57]. It seems plausible that an elevated requirement for an ssDNA exonuclease in this original reconstituted *E. coli* MMR system might have been driven by the choice of a mismatched DNA substrate containing a single GATC site. Under these conditions ssDNA exonuclease-dependent MMR could only be accomplished by the rare extended ssDNA exonuclease tracts or multiple sequential short ssDNA exonuclease events that were observed here (Fig. 5a, b).

The lack of extensive ssDNA exonuclease excision combined with a MUD$_{sat}$ (~1.7 kb) capable of displacing the vast majority of GATC→GATC segments strongly implicates an EcMutL–EcUvrD helicase-driven strand displacement mechanism for the majority of *E. coli* MMR excision. In support of this conclusion, we demonstrated efficient displacement of an ssDNA segment generated by dual strand breaks flanking a mismatch. The length of this segment was near the MUD (1.1 kb) for the EcUvrD concentration (20 nM) utilized in the ensemble MMR reaction, suggesting that most *E. coli* GATC→GATC segments should be easily removed by helicase-driven strand displacement. Moreover, the MUD$_{sat}$ observed here generally occurred within the doubling time of *E. coli* even though the smFS/smTIRF studies were performed at a reduced incubation temperature (23 ºC) as a result of instrument configuration.

Radman and Wagner first introduced the idea that *E. coli* MMR excision might occur between adjacent GATC sites in 1986[58]. A predominant unwinding-displacement mechanism between adjacent GATC→GATC EcMutH incision segments is extraordinarily similar to nucleotide excision repair (NER) that utilizes EcUvrD to efficiently remove a 12-mer ssDNA segment produced by EcUvrABC-induced incisions on either side of UV-damaged nucleotides[59]. The major difference between MMR and NER is the significantly greater processivity of EcMutL-tethered EcUvrD compared to EcUvrD alone.

The original *E. coli* exonuclease-dependent archetype appears similar to an exonuclease (EXOI) dependent pathway identified in *Saccharomyces cerevisiae* and reconstituted human biochemical systems[60–62]. The EXOI-dependent pathway contrasts a prominent exonuclease-independent MMR pathway also identified in *S. cerevisiae* that utilizes an intrinsic endonuclease activity found on the vast majority of MLH/PMS proteins, which is absent in EcMutL[2,60,63,64]. It is intriguing that the MLH/PMS endonuclease appears to introduce strand-specific breaks similar to the Dam/MutH system;[65] possibly suggesting an analogous exonuclease-independent strand displacement MMR mechanism[66].

The dynamic nature of the *E. coli* MMR components strongly supports the Molecular Switch/Sliding Clamp model as originally detailed for *E. coli* in 2003[6]. Small differences between this non-traditional model and the real-time images of MMR shown here and in previous studies[15] include the incision of hemimethylated GATC sites by the EcMutS–EcMutL/EcMutH complex for some distance surrounding the mismatch (Fig. 8a) and the detailed mechanics of EcMutL-tethered EcUvrD helicase unwinding (Fig. 8b)[6]. While the stochastic progressions of *E. coli* MMR[67] might seem inefficient, employing multiple randomly diffusing sliding clamps to tether components to the DNA appears to provide significant redundancy to the system. Interestingly, accomplishing excision by displacing the DNA between adjacent GATC→GATC incision segments introduces an additional stochastic layer to MMR, since repair will only be complete when the segment containing the mismatch is displaced and subsequently resynthesized (Fig. 8c–f). Until that happens redundant EcMutL–EcUvrD complexes may randomly remove adjacent GATC→GATC segments that do not contain the mismatch. For

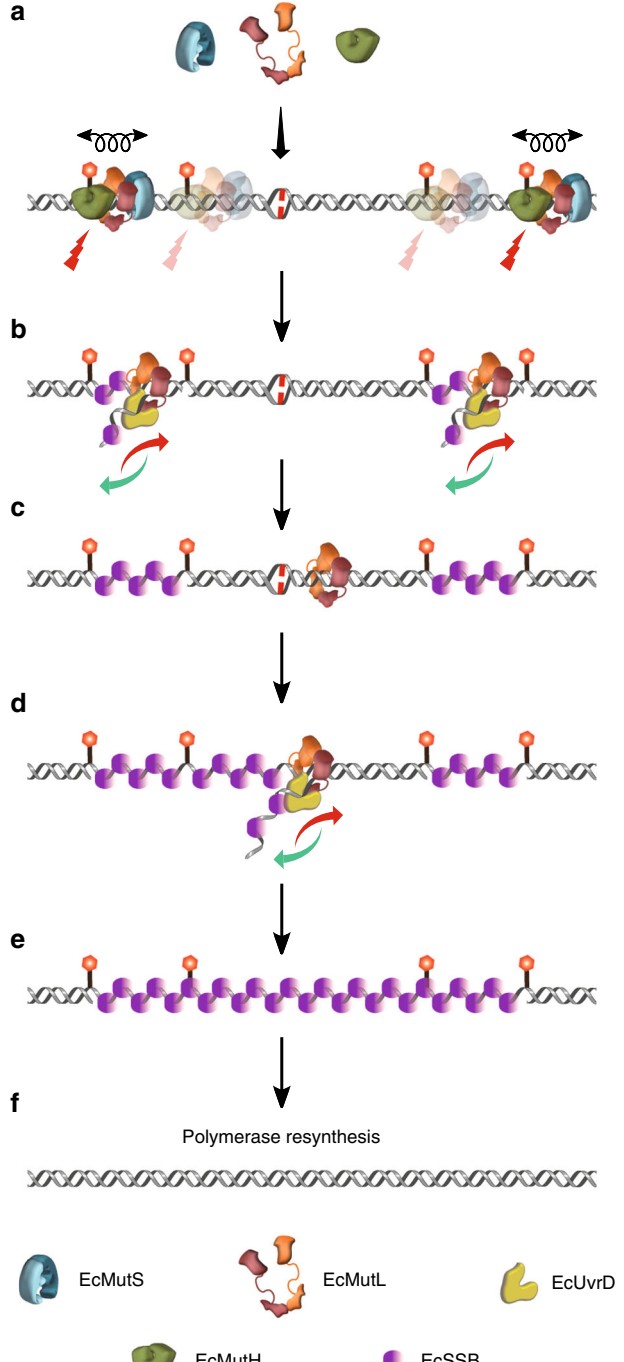

**Fig. 8** A complete model for strand specific excision by *E. coli* mismatch repair. **a** Cascading EcMutS (blue) and EcMutL (brown) clamps recruit EcMutH (green) to generate multiple strand scissions (red lightning bolts) on a mismatched DNA at hemimethylated GATC sites. **b** EcMutL captures EcUvrD (yellow) near an EcMutH strand scission tethering it to the mismatched DNA where it randomly unwinds (red arrow) and rezips (green arrow) the DNA that is alternately bound and released by EcSSB. **c** When the EcMutL–EcUvrD unwinding reaches an adjacent EcMutH GATC incision site, the intervening fragment is released creating an EcSSB bound gap. **d**, **e** While displacement of GATC→GATC may occur randomly between adjacent sites, MMR is not completed until the segment containing the mismatch is released. **f** The replicative DNA polymerase and ligase complete MMR by resynthesizing the gaps generated by EcMutL–EcUvrD strand displacement.

the small number of adjacent GATC→GATC incision-segments that are longer than the MUD_sat of EcMutL–EcUvrD, we envision either rare extended unwinding-rezipping events that exceed the MUD_sat, converging events from adjacent strand breaks or successive short ssDNA exonuclease events that produce a truncated ssDNA segment which may be eventually displaced by the MUD_sat. Alternatively, the loading of EcMutS, EcMutL, EcMutH, and EcUvrD might occur near the replication fork such that the segment containing the mismatch is more resourcefully removed; perhaps utilizing transient strand breaks associated with replication[68]. Taken as a whole it appears that every step of MMR is stochastic and that the evolutionary development of collaborating highly conserved EcMutS and EcMutL sliding clamps that are linked to the DNA ensures accurate strand-specific excision while making the process comparatively resistant to thermal disruption.

## Methods

**Plasmid construction, protein labeling, and purification**. The EcMutS, EcMutL, EcMutH, and EcRecJ proteins were purified and labeled utilizing the Hydrazinyl-Iso-Pictet-Spengler (HIPS) ligation method[15]. EcExoVII and EcSSB (unlabeled) were purchased from New England Biolabs or Thermo Fisher Scientific, respectively. The *E. coli uvrD*, *ssb*, and *exoI(xonA)* genes were amplified by PCR (Supplementary Table 1), digested with *XbaI*, and *XhoI* (for *UvrD*), *NdeI* and *EcoRI* (for *SSB*) or *NdeI* and *BamHI* (for *ExoI*), and inserted into pET-29a (Novagen) bacterial expression plasmid. Hexa-histidine (his_6) and Formylglycine Generating Enzyme (FGE) recognition hexa-amino acid sequence (LCTPSR; ald_6) were introduced onto the C-terminus of EcUvrD and EcExoI proteins. Two glycine residues separated the his_6 and ald_6 and these tags were separated from the MMR proteins by two serine residues. The order of these tags relative to the MMR gene is indicated (Supplementary Table 2). To label EcSSB, a single cysteine point mutation (A123C) was generated using the QuikChange site-directed mutagenesis kit (Stratagene)[69]. All the plasmid constructs were amplified in *E. coli* XL10 gold (Stratagene) and verified by DNA sequencing.

EcUvrD was expressed, labeled and purified by modification of a previous protocol[15,70]. Briefly, after co-transformation with the MtFGE and EcUvrD expression plasmid, a single colony of BL21 AI cell was diluted into 1 L of LB containing 50 μg/ml kanamycin and ampicillin. At OD_600 = 0.3, the growth temperature was decreased to 16 °C and the expression of MtFGE and EcUvrD was induced by addition of L-(+)-Arabinose (0.2% wt/vol) and IPTG (0.1 mM). Cells were collected after 16 h and resuspended in Freezing Buffer (25 mM Hepes pH 7.8, 300 mM NaCl, 10% glycerol and 20 mM imidazole). Cell pellets were frozen-thawed three-times and sonicated twice, followed by centrifuged at 41,000 rpm (Rotor: Ti 60 Beckman) for 1 h. The supernatants were then loaded on a Ni-NTA (Qiagen) column, washed with Buffer A (25 mM Hepes pH 7.8, 300 mM NaCl, 10 % glycerol and 20 mM imidazole) and eluted with a 20–200 mM Imidazole in Buffer A. Fractions containing EcUvrD proteins were pooled and dialyzed against labeling buffer (100 mM potassium-phosphate pH 7.0, 0.25 mM DTT, 300 mM NaCl) overnight. The protein fraction was then incubated with AF647-HiPS (Hydrazinyl-Iso-Pictet-Spengler) dye at 0 °C for 48 h. After labeling, EcUvrD proteins were diluted with 3 volume of Buffer B (25 mM Hepes pH 7.8, 1 mM DTT, 10% glycerol) and loaded onto a heparin column, washed with Buffer B plus 100 mM NaCl and eluted with a linear gradient of 0.1–1 M NaCl. EcUvrD-containing fractions were dialyzed against Storage Buffer (25 mM Hepes pH 7.8, 1 mM DTT, 0.1 mM EDTA, 150 mM NaCl, 20% glycerol,) and frozen at −80 °C.

EcSSB was expressed, labeled and purified by a modification of a previous protocol[71]. Briefly, after transformation with the EcSSB expression plasmid, a single colony of BL21 AI cell was diluted into 1 L of LB containing 50 μg/ml kanamycin. Expression of EcSSB was induced by addition of L-(+)-arabinose (0.2 % wt/vol) and IPTG (0.1 mM) at OD_600 = 0.3. Cells were collected after 5 h at 37 °C and resuspended in SSB lysis buffer (50 mM Tris-HCl pH 8.0, 200 mM NaCl, 15 mM spermidine trihydrochloride, 1 mM EDTA and 10% sucrose). Cell pellets were then incubated with 200 μg/mL lysozyme at 4 °C for 30 min and centrifuged at 17,000 × g for 80 min. Polymin P (final concentration of 0.4% wt/vol) was then added to the supernatant to precipitate the EcSSB. The Polymin P precipitant was collected by centrifugation at 6000 × g for 20 min, gently resuspended in TGE Buffer (50 mM Tris-HCl pH 8.0, 1 mM EDTA and 20% glycerol) containing 0.4 M NaCl, followed by addition of solid ammonium sulfate (150 g/L). The ammonium sulfate precipitant was centrifuged at 17,000 × g for 30 min and the pellet gently resuspended in TGE buffer containing 0.3 M NaCl followed by chromatography on a single-stranded DNA-cellulose column. The column was washed with TGE buffer containing 0.3 M NaCl and eluted with a linear gradient of 0.3–2 M NaCl. Fractions containing EcSSB proteins were pooled and dialyzed against SSB Labeling Buffer (50 mM Tris-HCl pH 8.0, 500 mM NaCl, 20% glycerol and 0.1 mM DTT) for overnight. Cy3-maleimide (Lumiprobe) was covalently conjugated to the single free cysteine of EcSSB at a 20-fold molar excess of Cy3 to protein. Excess Cy3 dye was removed by single-stranded DNA-cellulose chromatography, the Cy3-labeled

### Figure legend labels

EcMutS

EcMutL

EcUvrD

EcMutH

EcSSB

EcSSB fractions dialyzed against SSB Storage Buffer (50 mM Tris-HCl pH 8.0, 500 mM NaCl, 20 % glycerol and 1 mM DTT) and frozen at −80 °C.

After transformation with the EcExoI expression plasmid, a single colony of BL21 AI cell was diluted into 1 L of LB containing 50 µg/ml kanamycin. Expression of EcExoI was induced by addition of L-(+)-arabinose (0.2 % wt/vol) and IPTG (0.1 mM) at $OD_{600}$ = 0.3. Cells were collected after 3 h at 37 °C and resuspended in freezing buffer (25 mM Hepes pH 7.8, 300 mM NaCl, 10% glycerol and 20 mM imidazole). Cell pellets were frozen-thawed three times and sonicated twice, followed by centrifuged at 41,000 rpm (Rotor: Ti 60 Beckman) for 1 h. The supernatants were then loaded on a Ni-NTA (Qiagen) column, washed with Buffer A (25 mM Hepes pH 7.8, 300 mM NaCl, 10 % glycerol and 20 mM imidazole) and eluted with a 20–200 mM Imidazole in Buffer A. Fractions containing EcExoI (>95% purity) were pooled and dialyzed against Storage Buffer (25 mM Hepes pH 7.8, 1 mM DTT, 0.1 mM EDTA, 150 mM NaCl, 20% glycerol,) and frozen at −80 °C.

All proteins were expressed as the monomer concentration. The concentrations of unlabeled proteins were determined by absorbance spectrophotometry at 280 nm using the following extinction coefficients (EcUvrD, $\varepsilon_{280} = 106{,}208$ cm$^{-1}$ M$^{-1}$, EcSSB $\varepsilon_{280} = 28{,}023$ cm$^{-1}$ M$^{-1}$, EcExoI $\varepsilon_{280} = 75{,}205$ cm$^{-1}$ M$^{-1}$). The labeling efficiencies of AF647–EcUvrD monomer (38%) and Cy3-EcSSB monomer (45%) were determined by examining absorbance at 650 nm (AF647, $\varepsilon_{650} = 239{,}000$ cm$^{-1}$ M$^{-1}$) or 550 nm (Cy3, $\varepsilon_{550} = 150{,}000$ cm$^{-1}$ M$^{-1}$) and comparing the molar ratio between protein and fluorophore.

**Single-molecule imaging buffers and experimental conditions.** Single-molecule imaging Buffer C contains 20 mM Tris-HCl (pH 7.5), 5 mM $MgCl_2$, 100 mM NaCl, 0.1 mM DTT, 0.2 mg/mL acetylated BSA (Promega) and 0.0025% P-20 surfactant (GE healthcare). To minimize photoblinking and photobleaching, imaging buffer was supplemented with a photostability enhancing and oxygen scavenging cocktail containing saturated (~3 mM) Trolox and PCA/PCD oxygen scavenger system composed of PCA (1 mM) and PCD (10 nM)[72].

**Construction of 18.4-kb mismatched DNA with a single nick.** A plasmid containing two adjacent *BbvCI* sites was first treated with *Nb.BbvCI* (New England Biolabs), then followed by heating-reannealing-ligation with 1000 X oligo 1 to generate a permanent 3' nick on DNA (Supplementary Fig. 1; Supplementary Table 1; for 5' nick, *Nt.BbvCI* and oligo 2 were used instead). The resulting DNA was then digested by *BsaI* and separated on a 0.5% low melting agarose (Promega) gel. The 7 kb band was excised and treated with β-agarase (New England Biolabs) followed by isopropanol precipitation. Concurrently, λ phage DNA (3.2 nM, Thermo Scientific) was ligated with the oligo 3 and oligo 4 (800 nM; Supplementary Fig. 1; Supplementary Table 1) at room temperature (23 °C) overnight. Unligated oligonucleotides were removed using a 100 kDa Amicon filter (Millipore). The resulting λ DNA was then digested with *BsaI* at 37 °C for 3 h, ligated with the 7-kb DNA containing a permanent nick, 1000 X oligo 5 and oligo 6 (Supplementary Table 1) at 18 °C overnight. DNA ligation products were separated on a 0.5% low melting agarose (Promega) gel and the 18.4-kb band was excised and treated with β-agarase (New England Biolabs) followed by isopropanol precipitation. The purified DNA was resuspended in TE buffer (10 mM Tris-HCl, pH 7.5, 1 mM EDTA) and stored at −80 °C until use. For experiments including EcMutH (Fig. 1d), mismatched DNA (1 µg) was further incubated with 80 µM S-adenosylmethionine and 8 U of *Dam* methyltransferase (New England Biolabs) at 37 °C for 2 h in a 100 µL reaction, followed by inactivation of the enzyme at 65 °C for 15 min.

**Single molecule total internal reflection fluorescence microscopy.** All the single molecule total internal reflection fluorescence (smTIRF) imaging in this study were acquired on a custom-built prism-type TIRF microscope based on the Olympus microscope body IX71[15]. Fluorophores were excited using the laser lines (532 nm for green, 635 nm for red) in the smTIRF system. Image acquisition was performed using an EMCCD camera (ProEM Exelon512, Princeton Instruments) after splitting emissions by a Dual View optical setup (DV2, Photometrics). Micro-Manager image capture software was used to control the opening and closing of a shutter, which in turn controlled the laser excitation[73].

The 18.4-kb mismatched DNA (300 pM) in 300 µL T50 buffer (20 mM Tris-HCl, pH 7.5, 50 mM NaCl) was injected into the flow cell chamber and stretched by laminar flow (250 µL/min). The stretched DNA was anchored onto a neutravidin coated, PEG passivated quartz slide surface, and the unbound DNA was flushed by similar laminar flow.

To determine EcUvrD unwinding frequency, EcMutS (unlabeled, 100 nM), EcMutL (unlabeled, 100 nM), EcMutH (unlabeled) and AF647–EcUvrD (20 nM) in imaging buffer plus 2 mM ATP (unless stated otherwise) were introduced into the flow cell chamber and protein-DNA interactions were monitored in real-time in the absence of flow at ambient temperature. The DNA was stained with Syto 59 (1000 nM, Invitrogen) after recording.

To determine the interaction between EcMutL sliding clamp and EcUvrD, EcMutS (unlabeled, 10 nM) and AF555-EcMutL (20 nM) in imaging buffer plus 1 mM ATP were first co-injected. After 5 min the flow cell was flushed with imaging buffer containing AF647–EcUvrD (20–50 nM) plus 1 mM ATP and

protein-DNA interactions were monitored in real-time in the absence of flow at ambient temperature.

To detect the EcSSB coated ssDNA (Fig. 3d, e), EcMutS (unlabeled, 100 nM), EcMutL (unlabeled, 100 nM), AF647–EcUvrD (10 nM) and EcSSB (for 10 nM: 10 nM Cy3-EcSSB; for 200 nM: 30 nM Cy3-EcSSB plus 170 nM unlabeled EcSSB) in imaging buffer plus 2 mM ATP were introduced into the flow cell chamber and protein-DNA interactions were monitored in real-time in the absence of flow at ambient temperature.

To detect the extensive excision during MMR (Fig. 5c, d; Supplementary Fig. 6d), EcMutS (unlabeled, 100 nM), EcMutL (unlabeled, 100 nM), EcUvrD (unlabeled, 20 nM), EcSSB (30 nM Cy3-EcSSB plus 170 nM unlabeled EcSSB), EcExoI (20 nM), EcExoVII (0.1 U/µL, New England Biolabs) and EcRecJ (20 nM) in imaging buffer plus 2 mM ATP were introduced into the flow cell chamber and protein-DNA interactions were monitored in real-time in the absence of flow at ambient temperature.

**DNA substrate for single-molecule flow-stretching.** DNA substrate used for the single molecule flow-stretching (smFS) analysis was slightly modified by replacing the biotin oligo 6 with digoxigenin-labeled oligo 7 (Supplementary Fig. 1; Supplementary Table 1).

To construct the 22 kb long homoduplex DNA (Supplementary Fig. 3a), λ −phage DNA was digested by CsiI restriction enzyme (FastDigest, Thermo Scientific). The 22 kb fragment of λ−phage DNA was ligated with oligo 8, oligo 9 and oligo 10 (Supplementary Table 1) in molar ratio of 1:10:10:6 at 16 °C overnight. The resulting 22 kb duplex DNA was digested by Nt.BspQI (New England Biolabs) that generates five nicks on the 22 kb duplex DNA.

To construct a DNA with two strand breaks flanking the mismatch, a CRISPR/Cas9 system was used (Fig. 7c). An 18.4 kb mismatched-DNA in the absence of any nick was first constructed by a slightly modified protocol, where the nick generation step was skipped (Supplementary Fig. 1a). CRISPR RNA (crRNA 1 and crRNA 2; Supplementary Table 1) for each nick and trans-activating RNA (tracrRNA) were mixed and annealed in molar ratio of 1:1.5 to form single guide RNA (sgRNA) by cooling down from 90 °C to room temperature for 1 h with a thermal cycler (Applied Biosystems 2720 thermal cycler). The incubation of the mixture of sgRNA and Cas9 nickase (EnGen® Spy Cas9 Nickase, New England Biolabs) in 1:1 molar ratio at room temperature for 20 min formed RNA protein complex (RNP). The RNP was added to DNA substrates at a molar ratio of 10:1 to generate nicks, and then incubated overnight at 37 °C. To digest the RNP, RNase A (20 mg/ml, Sigma-Aldrich) was added to the reaction solution in a final concentration of 1 mg/ml for 15 min at 37 °C, while Proteinase K (20 mg/ml, Invitrogen) was added in a final concentration of 1 mg/ml for 30 min in 50 °C. 200 mM EDTA was added to the solution to stop the reaction. The resulting DNA was resuspended in TE buffer (10 mM Tris-HCl, pH 7.5, 1 mM EDTA) and stored at −80 °C.

**Single-molecule flow-stretching analysis.** All single-molecule flow-stretching experiments were performed in buffer D (20 mM Tris-HCl, pH 7.5, 5 mM $MgCl_2$ 125 mM NaCl, 0.1 mM DTT, 0.2 mg/ml BSA, 2 mM ATP) containing EcMutS (100 nM), EcMutL (100 nM), EcUvrD (20 nM, unless otherwise indicated), EcSSB (200 nM, unless otherwise indicated), and EcExoI (20 nM unless otherwise indicated) and/or EcExoVII (0.1 U/µL, New England BioLabs, unless otherwise indicated) and/or EcRecJ (20 nM, unless otherwise indicated). The identical flow-stretching analysis has been described previously[43,74]. The flow channel with $25.0 \times 3.0 \times 0.1$ mm dimension was built with a glass slide washed by acetone and a surface-passivated cover glass. The cover glass was functionalized with poly-ethylene glycol (PEG)-biotin and PEG (Laysan Bio) with a mass ratio of 1–100 to minimize the nonspecific binding of DNA substrates and proteins to its surface. To immobilize biotin labeled DNA substrates, streptavidin molecules (0.05 mg/ml, Sigma-Aldrich) in phosphate-buffered saline (PBS) buffer were incubated in the flow chamber for 10 m and then the free streptavidin molecules were washed out with a blocking buffer (20 mM Tri-HCl, pH 7.5, 2 mM EDTA, 50 mM NaCl, 0.0025% Tween 20 (v/v), 0.1 mg/ml BSA). The mismatched DNA substrates were attached to the cover glass surface of the flow chamber via a biotin-streptavidin linkage by flowing a DNA (~0.3 pM) in the blocking buffer for 10 m using a syringe pump (0.04 ml/m, Harvard apparatus). Free DNA molecules in solution were removed by stringently washing (~0.2 ml of blocking buffer). Antidigoxigenin antibody (antidigoxigenin Fab, Roche) coated super-paramagnetic (SPM) beads (2.8 µm in diameter, Invitrogen) were introduced to the flow chamber to bind to the immobilized DNA substrates containing a digoxigenin at the free-end. Prior to the addition of proteins, free SPM beads were removed by extensive washing (>2 ml of blocking buffer). A hydrodynamic force produced by laminar flow of the buffer was applied to a tethered SPM bead at ~2.2 pN[74]. A magnetic force generated by a ring shaped rare earth magnet was also applied to the SPM beads at ~1 pN to avoid nonspecific interactions between the bead and the surface[74]. The calculated net force acting on the SPM beads was 2.5 pN. The SPM bead linked to DNA was imaged using an optical microscope under a ×10 objective (N.A. = 0.40, Olympus). Images were recorded with a high−resolution CCD (RETIGA 2000R, Qimaging) using MetaVue (Molecular Devices) imaging software with a 1 s time resolution. The position of bead was determined using DiaTrack 3.03[43], and the data were analyzed by OriginPro8 (OriginLab) and Matlab R2016b (Mathworks). All the

experiments were carried out at room temperature (~23 °C) unless otherwise indicated.

**MMR complementation in vivo**. *E.coli* strains were kindly supplied by Patricia L. Foster (Indiana University) and were all derivatives of MG1655 (F- lambda- *ilvG-rfb*-50 *rph*-1). Mutation rates to rifampicin-resistance (Rif$^r$) were determined using at least nine independent colonies for each genotype[75]. Δ*uvrD* strains were co-transformed with EcUvrD expression plasmids (Supplementary Table 2) and pTARA plasmid (for T7 RNA polymerase expression, a gift from Kathleen Matthews, Addgene plasmid #31491)[76]. Single colonies were picked and grown for 24 h in the presence of 50 μg/mL Kanamycin, 35 μg/mL Chloramphenicol and 0.2% Arabinose. As controls, single colonies of *wild type* and Δ*uvrD* strains with pTARA plasmid were grown for 24 h in the presence of 35 μg/mL Chloramphenicol and 0.2% Arabinose. Dilutions of the cultures were plated on LB-Agar plates with or without 100 μg/mL rifampicin and allowed to grow overnight at 37 °C. The colonies on LB with or without rifampicin were counted and the mutation rates were determined by fluctuation analysis[77].

**Exonuclease activity analysis**. A 2038 nt ssDNA was generated by Guide-it™ Long ssDNA Production System (Takara) using oligo 11 and oligo 12 as primers. The exonuclease activities were examined using 15 nM of the 2038 nt ssDNA substrate incubated with 20 nM exonuclease (for EcExoI or EcRecJ) or 0.33 U/μL (for EcExoVII) in 30 μL reaction containing 20 mM Tris-HCl (pH 7.5), 5 mM MgCl$_2$, 125 mM NaCl and 0.1 mM DTT at indicated temperature for 1–30 min. Where indicated, 2 μM EcSSB (Thermo Fisher Scientific) or 500 nM EcUvrD plus 1 mM ATP were included in the exonuclease assay. The reactions were stopped by addition of 25 mM EDTA, 0.2% SDS, 0.67 mg/mL Proteinase K (Denville) and incubated at 50 °C for 1 h. Resulting DNA was resolved on a 1% agarose gel, scanned on a Sapphire Biomolecular Imager (Azure Biosystems) and quantified by ImageQuant software.

**Data analysis of TIRF imaging**. For studies involving AF555-EcMutL and AF647–EcUvrD, fluorescent molecules in two channels were co-localized using a custom written MATLAB script. Kymographs were generated along the DNA by a kymograph plugin in ImageJ (J.Rietdorf and A. Steiz, EMBL Heidelberg)[15]. AF647–EcUvrD was tracked by a custom written MATLAB script, in which the particle intensities were fit to a two-dimensional Gaussian function to obtain their positions with sub-pixel resolution. AF647–EcUvrD molecules with a minimum lifetime of 10 s (1 s frame rate) and a minimum DNA movement of 333 nm (2 pixels, unidirectionally) were counted as unwinding events ($N_{un}$). Following the real-time single molecule recording, the number of DNA molecules ($N_{DNA}$) was determined by Syto 59 staining. The frequencies of EcUvrD unwinding ($F_{un}$) were calculated using the following equations:

$$F_{un} = \frac{N_{un}}{N_{DNA}} \qquad (1)$$

All single molecule frequency studies were performed at least two separate times.

To measure the exonuclease excision during MMR, Cy3-EcSSB with a minimum movement of 333 nm were examined. Particles were tracked and fluorescent intensities were plotted. Particles were grouped into two categories based on their remaining fluorescent intensities after unwinding-rezipping: non-excision event and excision event (see Fig. 5c, d).

To acquire the initiate position of EcMutL–EcUvrD co-localization prior to unwinding (Supplementary Fig. 2c), the left ($P_L$) and right ($P_R$) positions of the DNA end were first estimated by tracking the end-to-end diffusion of the EcMutL sliding clamp. Then the first frame of co-localization was determined as the initial position ($P_I$) of unwinding. The positions were then converted to lengths in bp with Eq. 2, where 18,378 bp is the length of the mismatched DNA. A 1000 bp (~2 pixels) binning size was used to construct the position histograms.

$$18,378 \text{ bp} \times \frac{(P_I - P_L)}{(P_R - P_L)} \qquad (2)$$

**Analysis of smFS data**. To determine the spatial resolution of the smFS system, we selected 200 positions of the SPM bead in the longitudinal direction to the flow-stretching force prior to an unwinding event in a time trajectory and obtained the standard deviation value (20 ± 2 nm, $n = 10$) by Gaussian fit to the histogram of the 200 data points. The standard deviation is the minimum distance that can be resolved.

To convert an SPM bead position at the nanometer scale into the number of nucleotides unwound by EcUvrD and excised by EcExoI, we adopted our previously validated conversion factors[43]. Briefly, the end-to-end distances of dsDNA, ssDNA with the same amounts of nucleotides of the dsDNA, and ssDNA in the presence of EcSSB were measured under a constant extension force (2.5 pN). The difference between the lengths of ssDNA, SSB-bound ssDNA, and dsDNA provided the conversion factors. For these studies, the dsDNA → ssDNA conversion factor is 3.2 nt/nm; the dsDNA → SSB-bound ssDNA conversion factor is 4.2 nt/nm.

The DNA-tethered SPM beads with a minimum movement of 100 nm were counted as unwinding events ($N_{un}$). Multiple events occurring on a single DNA molecule were counted as a single unwinding event. The number of DNA molecules ($N_{DNA}$) in a single image was counted by flow reversal before the MMR protein injection[43]. The frequency of EcUvrD unwinding ($F_{un}$) was calculated using Eq. 1.

The unwinding events were further grouped in two different categories (Fig. 3c): unwound (DNA that remained unwound and/or included a DSB during the 25 min imaging period; see: Fig. 2b, d as examples) and restored (DNA that was unwound and was followed by rezippings in which the bead returned to its original position of fully duplex DNA; see: Fig. 3b as an example). We counted it as a single restored event when multiple rezipping events occurred on the same DNA molecule.

To establish the average excision length at various conditions (Fig. 5b and Fig. 6a), the difference between the average position of the SPM bead for 50 s before DNA unwinding and the average position of the SPM bead for 50 s after DNA rezipping was determined and plotted as a single exponential decay (Supplementary Fig. 5b). We excluded traces that displayed a drift that resulted in a position change of more than 7.5 bp for 50 s (9% of the total events). The exonuclease-dependent excision lengths (Fig. 6b; Supplementary Table 3) were obtained by $m_{exo} - m_{no-exo}$, where $m_{exo}$ ($\pm \delta a$) and $m_{no-exo}$ ($\pm \delta b$) are the mean values (±s.e.) of the excision lengths in the presence and absence of exonuclease, respectively. The s.e. of the resulting excision lengths were calculated by $\sqrt{(\delta a)^2 + (\delta b)^2}$. The density distribution of ssDNA exposure dwell time versus excision length (Fig. 6d) was plotted using 2D kernel density function in Origin.

**Binning method**. All binned histograms were produced by automatically splitting the data range into bins of equal size by using the Origin program.

**Distance calculation between adjacent GATC sites in *E. coli***. The *E. coli* K-12 MG1655 genome (NCBI: NC_000913.3) was used to search for GATC sites from which the distance between each site was calculated by a MATLAB script.

**Reporting summary**. Further information on research design is available in the Nature Research Reporting Summary linked to this article.

## Data availability
The data that support the findings of this study are available from the corresponding author upon reasonable request. The source data underlying Figs. 1d, 2c, 2f, 2g, 3c, 4a, 4b, 5b, 5d, 6a, 7a, 7e and Supplementary Figs 1c, 1d, 2c, 3d, 4a–e, 5b and 6a–c are provided as a Source Data file.

## Code availability
The MATLAB scripts for particle co-localization and tracking are available from the corresponding author upon request.

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

## Acknowledgements

We would like to thank our laboratory colleagues for many helpful insights and discussions. This work was supported by the Global Research Lab Program through the NRF of Korea funded by the Ministry of Science and ICT (2017K1A1A2013241; J.-B.L.) and NIH grants CA67007 and GM129764 (R.F.).

## Author contributions

J.L., R.L., B.M.B., J-B.L. and R.F. designed the experiments; J.L., B.M.B., J.H. and J.A.L., performed genetic analysis; J.L. and B.M.B. purified and labeled the proteins; J.L., R.L., B.M.B., K.Y. and J.H. performed the single molecule studies; J.L., R.L., B.M.B., J-B.L. and R.F. analyzed the data; J.A.L. developed the Matlab script for particle localization. J.L., R.L., B.M.B., J-B.L. and R.F. wrote the paper and all authors participated in critical discussions.

## Competing interests

The authors declare no competing interests.

## Additional information

**Supplementary information** is avaliable for this paper at https://doi.org/10.1038/s41467-019-13191-5.

