## [Peer Review File · Nature Communications]

Reviewer #1 (Remarks to the Author):

I have attached my comments as a Report.pdf file.

Report

Liu et al. have performed single molecule and bulk biochemical measurements to study *E. coli* DNA mismatch repair (MMR). The DNA constructs used for these measurements have a single 3' nick on a GATC site and a single mismatch 4.2 kb away from this nick. Using fluorescently labeled proteins, binding of MutL and MutL/MutS clamps around the nick site and recruiting and tethering of UvrD to these clamps have been demonstrated. In addition, unwinding and rezipping of double stranded DNA (dsDNA) by UvrD under different assay conditions has been demonstrated in these fluorescence measurements. A DNA construct that is stretched and tethered on both ends has been used for these measurements. The authors have also used flow stretching (FS) method to create essentially linear DNA constructs. For these measurements, one end of the DNA construct is tethered to the surface while the other end is conjugated to a superparamagnetic (SPM) bead. Under the flow conditions used in the study, ~2 pN force is applied to the SPM bead, which is also lifted up by a circular magnet to prevent it from sticking to the surface. In such a linearized DNA construct, unwinding of dsDNA results in contraction of the overall construct since single stranded DNA (ssDNA) needs greater force (due to smaller persistence length and higher entropy) to be stretched compared to dsDNA. UvrD unwinding activity has been quantified under different assay conditions by tracking the bead position and converting the contraction in contour length to number of unwound basepairs (or number of basepairs bound by EcSSB).

Using these concepts, the authors have compared maximum unwinding distance for different concentrations of UvrD in the presence and absence of MutL/MutS and varying concentrations of EcSSB. The authors have also quantified the level of exonuclease activity by the three canonical MMR ssDNA exonucleases. To accomplish this, the authors monitored whether a stretched DNA construct that is unwound by MutL/MutS-UvrD complex returns to its original position. A permanent change in the position is taken as evidence for exonuclease activity that degrades a segment of ssDNA and permanently prevents dsDNA formation.

The topic and questions addressed by the authors are of great significance. Fishel & Lee laboratories have published a respectable number of articles on various aspects of MMR in the last 5-6 years and this manuscript is part of these efforts where another piece of the mechanism is investigated: (i) modulation of processivity of UvrD by MutL/MutS clamps and EcSSB and (ii) Potential significance of UvrD catalyzed unwinding (rather than an exonuclease based digestion) in removing the strand with a mismatch. The assays used for these studies are well established and refined by the two laboratories to study proteins involved in *E.coli* MMR. The article was clear and easy to read. The number of molecules that constitute each histogram, the underlying uncertainties and how various quantities and uncertainties are calculated are explicitly given in the manuscript.

I have provided my comments for the significant conclusions of the study below:

(i) The EcMutL sliding clamp tethers EcUvrD to the mismatched DNA and increases its unwinding processivity: Dimerization and tethering to EcMutL increases UvrD processivity from ~20bp to over 9 kbp in this assay. This aspect of the work is consistent with previous reports (Yamaguchi et al. JBC 1998) however, the extend of enhanced processivity is significantly more than a recent report by Lohman lab that reported only a 2-3 fold increase in UvrD processivity due to MutL (Ordabayev *et al.* JMB, 2018). The authors might consider commenting on this difference. Ordabayev *et al.* also suggested that MutL may be moving along with UvrD, as would be expected from tethering model proposed in this study.

(ii) EcSSB restrains the extent of EcMutL-EcUvrD strand displacement: The authors demonstrate that high concentrations of EcSSB (200 nM) result in significantly reduced strand displacement. They propose this being due to high concentrations of EcSSB preventing UvrD dimerization, which is required for enhanced processivity.

This conclusion is fundamentally different from what is known about eukaryotic ssDNA binding protein RPA as RPA is known to dramatically increase the processivity various helicases such as BLM and WRN. However, the interpretation of the authors (reduced UvrD processivity due to inhibition of dimerization) is reasonable as low UvrD concentrations (1 nM) also showed significantly reduced MUD. Even though this mechanism would provide a way to regulate helicase activity, it would also reduce the likelihood of UvrD to displace the mismatched strand by processively unwinding the entire segment between two nicked GATC sites. The authors suggest that this is not a significant problem since the UvrD processivity they observe in the presence of 200 nM ECSSB (1.1 kbp) is still greater than the separation between vast majority of GATC sites. The authors should comment on whether the reduced processivity they observe at 200 nM EcSSB and 20 nM UvrD would be representative of cellular concentrations. In other words, is the processivity expected to be at a minimum when EcSSB:UvrD ratio is 10:1 as in this case or is this ratio higher/lower in cellular context? Is there reason to expect the processivity of UvrD would be reduced to a level that would make the proposed model based on UvrD catalyzed strand displacement unlikely?

In this context, the authors have also observed significantly higher levels of coating of ssDNA in the presence of 10 nM EcSSB compared to 200 nM EcSSB. This has similarities with reports on RPA where higher levels of exchange (dissociation from ssDNA curtain) were observed at higher concentrations of RPA (Gibb...Greene, PlosOne 2014). The authors might consider discussing this result in that context.

(iii) EcUvrD unwinding is rarely accompanied by exonuclease digestion: A widely accepted mechanism of MMR is that the newly synthesized strand that has a mismatch is degraded by exonucleases. The authors have cited some of the earlier work that demonstrated rapid degradation of long stretches of ssDNA (~2 knt) by ExoI in the presence of EcSSB. In contrast to these results, the authors have very rarely observed measurable exonuclease digestion and have proposed strand displacement between adjacent nicked GATC sites by UvrD to be the primary mechanism of removing the strand that has a mismatch. This is probably the most important conclusion of this study, and is potentially very significant. However, in its current form, it remains speculative. The authors should provide a direct evidence for this mechanism rather than leave it as an implication of their observations. One possible way could be designing a DNA construct with two GATC sites that are several hundred bp apart (average separation between adjacent GATC sites) and demonstrating the removal of the strand that has a mismatch between these sites. The length of that strand should be well defined and might be detectable in either single molecule assays or in gel electrophoresis at high enough concentrations. It may be possible to stop digestion of the free strand by exonucleases by stopping the reaction at different time points.

Minor Comments

1- Line 113-114: How do the authors know directed motion starts near the ssDNA break? It appears to start 1-2 micrometers away, which is comparable to the total available length on the 3' side of the break.

2- Lines 124-127: It is not clear why the described scenario would result in a double strand break. Can the authors clarify?

3- Lines 164-167: How would the histogram in Fig. 2f look like for UvrD alone when it is present at high enough concentrations to form dimers? The authors suggest MutL increases the frequency of unwinding events at the expense of re-zipping events. To put this statement in better context the same data as in Fig 2f should be shown when MutL is not present.

Also, there is an obvious skew in the distributions in Fig 2f (many of the other such peaks), which might be better fit with a Lorentzian, rather than a Gaussian and result in smaller uncertainties due to fitting.

4- Line 171: “bp/s” is used in Fig. 2f but “nt/s” used in the manuscript text. bp/s is probably more appropriate when describing unwinding.

5- Line 181: Image quality in Figure 3a not very good, especially the merged image. It could be an issue with converting to .pdf but a higher resolution image in the published version would help.

6- Line 185: I probably do not understand this correctly but it is not clear how the frequency of Restored be greater than those Unwound in Figure 3c at 200 nM and 1000 nM SSB. Any clarification?

7- Line 184-187: *“The majority of events in the absence or at sub-saturating concentrations of EcSSB resulted in very long tracts of unwound DNA (Fig. 2b) that rarely re-zipped to restore the original length and often terminated in a DSB (<10 nM EcSSB; Fig. 3c). However, at saturating EcSSB concentrations the majority of the mismatched DNA molecules were re-zipped to restore their original length (>200 nM EcSSB; Fig. 3b,c).”* Is there an intuitive way to understand why this happens to be the case? Aggregation of EcSSB at 10 nM but not 200 nM does not seem to be an obvious explanation of this. At saturating SSB concentrations, strand switching of UvrD should somehow be prevented but it is not clear how this would come from the different types of SSB filaments observed in the two cases. It should also be noted the frequency of re-zipping events is higher in the presence of SSB compared to its absence (Line 195-196).

8- Line 213-215: *“Taken together, we conclude that EcSSB regulates extensive unwinding by excluding secondary EcUvrD binding and/or multimerization on previously displaced ssDNA regions (Fig. 4g).”* In such a case, should it (exclusion of UvrD dimerization) not be more significant at 10 nM SSB (compared to 200 nM SSB) where more SSB coats the ssDNA? As should be clear from the last couple comments, the meaning of the fluorescence images of 10 nM and 200 nM SSB was not clear for this reviewer.

9- Line 252-253: This conclusion is based on the assumption that if the SPM length does not go back to its original position, the ssDNA must be excised. However this could also happen if re-zipping is prevented, maybe by coating the template or the displaced strand by one of the involved proteins that prevents re-zipping. Is there a way to distinguish between the two scenarios?

10- Line 280-281: If UvrD competes with MutH, how would the authors explain formation of MutS/MutL-MutH search complex? Would competition with vastly more abundant UvrD make this very unlikely and inefficient?

11- Line 285-287: Would it not be more efficient if the interaction domains did not overlap to simultaneously conscript both MutH and UvrD, since they would be not exclude each other?

12- Line 356-361: Not having a mechanism to bias the helicase towards the mismatch sounds very inefficient. In the absence of a bias towards the mismatch, the wrong segment will be displaced and needs to be resynthesized half of the time while the segment that has a mismatch will not be readily repaired.

Reviewer #2 (Remarks to the Author):

Based on a quite extensive set of genetic and biochemical evidence, the paradigm for the excision step in DNA mismatch repair (MMR) in *Escherichia coli* is that it is executed by any of a set of four exonucleases, possibly in conjunction with the helicase UvrD (MutU). In this manuscript the authors use two single-molecule biophysics assays (smTIRF and smFS) to challenge this paradigm. Essentially, the claim is that the exonucleases only serve to prevent resealing of the nick introduced by MutH, whereas the excision step, between adjacent nicks, is performed by the MutL/UvrD helicase heterodimer alone.

While the biophysical results largely support this hypothesis, it should be emphasized that this represents a reductionist approach. For instance, it is not sure whether the stoichiometry of the different proteins represents that in vivo. Unfortunately, there are no genetic or biochemical experiments provided in the manuscript to support the conclusion (other than reinterpretation of existing literature). For these reasons I feel that the strong conclusions that the authors draw require further support from additional in vivo and in vitro experiments, before the model presented here will replace the current paradigm. After all, strong claims require strong evidence.

Specific issues:

1. The Introduction claims that (at least in *E. coli*) MMR employs extremely stable MutS/MutL clamps. There is, as far as I know, no in vivo evidence for this. In this respect it is relevant to note that in eukaryotes, the MutS/MutL paralogs do not colocalize in vivo during MMR (Hombauer and Kolodner). It is also relevant to note in the manuscript DAM/MutH-dependent MMR is an idiosyncrasy of in *E. coli* and related gram-negative bacteria, that in mammals no helicase has been found to be involved and it is believed that ExoI removes the mismatched strand by sequential reloading (eukaryotes possess an additional, ExoI-independent, subpathway). Therefore, even in case the mechanism described here is true, it by no means should serve as a paradigm for MMR in general.
2. The use of a single mismatched substrate with a single (introduced) nick at the 3' of the mismatch represents a limited approach that might lead to artifacts. Thus, the substrate used for both assays is an 18,4 kbp molecule with a single mismatch and at its 3' side, at 4,2 kbp a single mismatch. This hardly is a physiological substrate as in vivo, the nick(s) in the large majority of cases the nick(s) will be within a few hundred nucleotides from the mismatch. Furthermore, in vivo, there will be nicks on each side of the mismatch.
3. In many of the substrate molecules there appear to be additional (inadvertent) nicks as evidenced by the occurrence of broken molecules following UvrD activity (indicating that the nicks are at opposite strands). This raises the possibility that many of the entry sites of UvrD (or of ExoI) are at these inadvertent nicks, rather than at the planned (3') nick. How can the authors be sure that the measured excision tracts have not been initiated at those inadvertent nicks? And why do these inadvertent nicks appear to be at the same position in all cases?
4. It would be reassuring to test excision at a matched substrate, to exclude artifacts caused by homoduplex binding of MutS. Other controls I would like to see is the use of a substrate with a nick at both sides of the mismatch (which would lead to MMR), and a substrate without a nick (which would

be refractory to excision). These would be very useful controls and would also support the hypothesis that *uvrD* removes the tract between those nicks.

5. Another experiment that would support the major hypothesis of the paper is to use a biochemical MMR assay (aka the Modrich assay), with a 5' nick, a 3' nick and nicks at both sides of the mismatch, and purified proteins. This can be performed using wild type and catalytically dead proteins (other than the MutL ATPase mutant). This also allows the mapping of excision tracts (as has been done by Radman using *Xenopus* oocyte extracts).

6. In case *UvrD* would be the enzyme performing excision in MMR, MMR would not be possible, or greatly reduced, when only a 5' nick is present (given the 3' to 5' polarity of the helicase). This is easily testable using the Modrich assay.

7. In relation to the previous points, have the smFS experiments been performed only with the 3' nick? I would like to see experiments also with a 5' nick (see also point 8).

8. ssDNA resulting from excision will probably trigger rapid SSB binding *in vivo*. *In vitro*, SSB appears to result in an increase in re-zipping events by *UvrD*. This appears counterintuitive as it will hamper excision. Is sliding of the MutS/MutL/*UvrD* clamp inhibited by SSB bound to ssDNA, introduced by unwanted excision at a 5' nick? In case this is true, did the authors consider the possibility that *in vivo* this might act to prevent unwanted *UvrD*-mediated excision events at a 5' nick?

9. The authors demonstrate that *ExoI* only has a very poor processivity in the presence of *UvrD* and hardly competes away the helicase from MutL. This is taken as primary evidence that excision is not performed by exonucleases. However, it cannot be excluded that *in vivo*, *ExoI* processivity is enhanced more by reloading of the MutL-*ExoI* complex than *UvrD* processivity (as is the case in mammalian MMR; Guo-Min Li, 2011).

10. The authors suggest that MutL-exonuclease activity only serves to prevent unwanted religation of the nick. But if *UvrD* binds so much better to MutL than *ExoI*, the nick may never 'see' a MutL-*ExoI* complex, how do the authors then envisage this?

Reviewer #3 (Remarks to the Author):

Liu et al

In this study, the authors perform a detailed series of single molecule experiments primarily examining the interactions between MutS, MutL and *UvrD* on mispaired DNA molecules and to a lesser extent examining how SSB and ssDNA exonucleases alter these interactions. Building on prior work, this study shows that in the presence of MutS and a Mismatch, MutL clamps and *UvrD* co-assemble to form a complex and are able to unwind DNA from a nick for a considerable distance which can extend to up to 2 to 3 KB (average ~1.1 kb) in the presence of high levels of SSB, that also stimulates reannealing (re-zipping) of the unwound strands behind the *UvrD*-MutL complex. They also provide some evidence that when any 1 of 3 ssDNA exonucleases implicated in MMR is present, a small proportion of the substrate is extensively resected and most of the substrate is only resected to a small extent. For the most part, the studies appear well done and the results justify the conclusions. Importantly, the key result demonstrating the MutS and mismatch dependent formation of a MutL-*UvrD* complex that can then unwind sufficiently long regions of DNA from nicks at hemimethylated GATC sites adjacent to a mismatch to excise the mismatch in the absence of an exonuclease has very intriguing implications for our understanding on MMR mechanisms that have generally been much more poorly

established that most investigators appreciate. More simply put, this study for the first time demonstrates a viable exo-independent mechanism for *E. coli* MMR. Thus, this is an interesting and thought-provoking paper that clearly advances the MMR field.

I do have some general concerns about this study/paper that need to be addressed. First, the results section is written in a very terse way that to some extent only single molecule biochemists will be able to understand even though the key audience is everyone else in the DNA repair field. The authors need to much more clearly explain how they draw the conclusions they do from each experiment they perform and they need to pay more attention to details such as how many molecules are actually examined in each experiment, which generally doesn't appear to be stated in any figure legend. This general criticism remains even though an extensive methods section is provided. Given the lack of length limitations, some additional space could be devoted to an expanded results section. Second, the analysis of the ssDNA exonucleases is not as rigorous as the rest of the analysis. This is because the unwinding/rezipping assay is only an indirect measurement of ssDNA. One wonders if this aspect could be strengthened if the ssDNA regions could in some way be stained by fluorescent SSB, possibly added after termination of the unwinding-exo reaction. Note, however, the exo aspects of the paper are the least important and their limitations could simply be explained better; some additional analysis would be preferable. Third, the discussion could be more compact and clearer. Examples include: a) reducing the discussion of NER to one sentence to point NER out as an example of a similar dual excision based mechanism; b) an expanded and clearer discussion of the earlier biochemical reconstitution studies and how they would have missed an exo independent reaction possibly stating a need to revisit bulk biochemical reconstitution studies; c) an expanded discussion of the similarities of the exo dependent and independent mechanisms proposed here to that well established in yeast; and d) deletion of the last paragraph that doesn't add much to the main point and may be misleading in that one might expect MutL-UvrD complexes to be preferentially assembled near the mismatch due to the involvement of MutS and therefore preferentially excise mispairs, something that isn't discussed.

Specific more minor comments

P2, L23-25. What does terrestrial biology have to do with this paper? And also, exo independent MMR mechanisms have clearly been published so this isn't the only paradigm.

P2, 40-42. It should be noted that this is a mechanism proposed from in vitro reconstitution studies. It should also be possible to note that this isn't the only mechanism known.

P3, L48. Given that non-bacterial MMR is barely mentioned in this paper, this sentence should really just focus on MutS and not mention homologs.

P5, L108. "These labeling conditions...." Should be restated in a way to make it clearer that it is those conditions used in Refs 31-34.

P5, L123 (and elsewhere). The authors never clearly explain why they think a DSB is formed or how they prove this point.

P8, L181. In regard to "little if any detectable movement", it would be useful if the authors were to provide a figure in which data showing movement and data showing lack of movement are shown side by side to allow the reader to better appreciate the two states.

P9, L213 (and elsewhere; e.g., P12, L288). I don't like the use of "regulates" as it implies an active process that occurs in response to a stimulus. What the authors mean is that the presence of SSB alters UvrD unwinding. SSB is constitutively present in cells and unless there is evidence for some

condition that alters SSB levels to change repair biochemistry, "alters" is a better word.

P11, L253. Delete "historical". Its really all prior reconstitution studies.

P13, L319. Delete "hypothetically" and substitute "only".

P13, L321-324. Delete this sentence. It doesn't add anything.

P14, L345-347. This is a mis-statement of what Ref 60 as well as multiple yeast papers (not cited) showed. Not only are the nicks seen broadly distributed along the nicked strand, none of the studies could detect more than 1 nick per strand even if more than one nick was present. The only study I know of in eukaryotes implying close nicks/short excision tracts is Muster-Nassal 1986 that observed very short mispair excision tracts in vitro, albeit using a rather crude assay.

P18, L443. The abbreviation "Nb" is not defined.

Supplementary Table 2. Shouldn't the mutation rates column be indicated as either "Mutation rates x 10e8" or "Mutation rates (10e-8)

RESPONSE TO SPECIFIC REFEREE COMMENTS:

Reviewer #1

Response to the significant conclusions of the study

(i) We elected not to comment on any differences between Ordabayev et al., (*JMB* 430:4260, 2018) compared to our work since this group used very short oligonucleotides and only included EcMutL and EcUvrD in the unwinding reactions (no EcMutS or a mismatch). In addition, this group performed their studies under very low (30-40 mM) ionic conditions. We have previously demonstrated that EcMutL binds nonspecifically to DNA at ionic conditions below 100 mM and does not form a sliding clamp that is essential for MMR (*PLoS One* 5:e15496, 2010; *Nature* 539:583, 2016). Since the physiological significance of these observations remains unclear, we are unable to make any constructive comparisons with our current studies. In contrast, Yamaguchi et al., (*JBC* 273:9197,1998) constructed a mismatched DNA substrate with dual strand breaks around a mismatch and included EcMutS with EcMutL and EcUvrD. However, these studies were also performed under very low ionic condition and without EcSSB. Nevertheless, we have now noted this paper as similar to our observations with some important variances.

(ii) Throughout the manuscript we have been very careful to not address the multimeric state of EcUvrD since our data is unable to unambiguously resolve this somewhat controversial topic. While, we agree that some aspects of our observations seem to suggest different multimeric

states of EcUvrD, our system lacks the resolution to definitively resolve this issue.

We have found that 200 nM EcSSB is saturating for the *in vitro* single molecule systems described in our manuscript (see: Fig. 3c). We have also examined up to 100 nM EcUvrD, where 20 nM is saturating for the ssDNA exonuclease studies (see: Fig. 6b) and 20-100 nM appears saturating for the MUD analysis (see: Fig. 4d). While the cellular concentrations of EcSSB and EcUvrD are approximately 20-fold higher, $\sim 5 \mu\text{M}$ and $\sim 3 \mu\text{M}$ respectively, the ratio of these proteins in our saturated *in vitro* single molecule imaging systems appears nearly identical to that found *in vivo*. Thus, there appears to be no discernible cellular conditions where EcUvrD or EcSSB might become limiting, and which could make the strand displacement model for *E. coli* MMR “unlikely”.

Finally, the biochemical cause for the differences in EcSSB coated tract lengths at 10 nM and 200 nM reflects the ssDNA tract length following EcUvrD unwinding and/or re-zipping [also see 8) below]. Thus, at low EcSSB (10 nM) the coated tracts are not saturated (presumably leaving naked ssDNA gaps) and appear experimentally similar to the absence of EcSSB where multiple EcUvrD molecules may be loaded and combine to promote long unwound regions. In contrast, at saturating EcSSB the coated tract length appear short and ultimately disappear, reflecting significantly more re-zipping that restores the substrate to fully duplex DNA (Fig. 3c). While these observations may appear analogous to other single molecule observations utilizing the eukaryotic single-stranded binding protein RPA, the release of the EcSSB from the mismatched DNA shown here is accompanied by catalytic EcUvrD re-zipping. These observations are consistent with an active enzymatic mechanism that displaces EcSSB, rather than the apparently passive mechanics described by Greene and colleagues (*PLoS One*, 2014).

(iii) The “earlier work” referred to by Referee #1 was actually presented in Suppl. Fig. 4b-d of the original submission (now Suppl. Fig. 6a-c). Nevertheless, both Referee #1 and #2 suggested we examine the repair of a DNA substrate containing defined ssDNA scissions flanking the mismatch. This data is now shown in Fig. 7 and demonstrates that the majority of excision tracts uniquely displace the DNA strand between these defined ssDNA scissions as predicted by the EcUvrD strand displacement MMR model.

Minor Comments

1) We have now analyzed more molecules for Suppl. Fig. 2c. The results show that the majority of initial EcMutL-EcUvrD interactions events occur within 200-400 nm of the defined strand break (Supplementary Fig. 2c). This spatial localization is consistent with the smTIRF instrument that is by nature diffraction-limited.

2) Unfortunately, it is nearly impossible to maintain a fully intact 18 kb mismatch-containing substrate over the course of construction, handling and single molecule imaging analysis. In our experience, these λ -based DNA substrates may randomly acquire a few ssDNA scissions as a result of this construction/handling. If an EcMutL-EcUvrD unwinding tract encounters one of these random strand breaks located on the opposite strand from the initiating ssDNA break, it will result in a DSB. Alternatively, we note that most of the DSBs occur following very long unwinding tracts, which could implicate some mechanical stresses in the process. It is important to note that the formation of DSBs is rare in the presence of EcSSB, where EcUvrD re-zips most of the molecules restoring them to a duplex DNA form.

3) Our inability to resolve multimeric forms of EcUvrD is addressed above. In the absence of EcMutL or in the presence of an ATP-binding deficient mutant of EcMutL [EcMutL(R95F)] that is unable to form a sliding clamp, we observe no DNA unwinding (Fig. 1d; Fig. 2c). Thus, with the

smTIRF and smFS systems we are unable to examine the frequency of unwinding and/or re-zipping in the absence of EcMutL. However, previous studies from the Lohman and Ha groups have shown that the frequency of EcUvrD unwinding and re-zipping (absence of EcMutL) on very short oligonucleotides is identical (*Science* **348**:6232, 2015).

We appreciate the suggestion by Referee #1 to fit Fig. 2f to a Lorenzian distribution that might better account for the “tails” associated with the unwinding and re-zipping rates. However, like the Gaussian distribution the Lorenzian distribution is symmetric, which would place the left “unwinding” tail into the negative “re-zipping” class. Since this did not make physical sense and the mean of both distributions is well within the standard-of-deviation of either distribution, we elected to use the Gaussian.

4) We have changed the notation to bp/s.

5) The image in Fig. 3a is of general high quality.

6) We have re-crafted the text and Fig. 3c to better define “unwound” and “restored” EcMutL-EcUvrD events.

7) See 2) and 6) above. We have included additional text on how an EcUvrD unwinding tract might encounter a sporadic ssDNA break on the opposite strand, which could result in a DSB. Importantly, at saturating EcSSB re-zipping that restores the original substrate length is significantly more frequent. While it is likely that strand switching is ultimately responsible for re-zipping as indicated by previous studies (*Proc Nat Acad Sci USA*. 101:6439, 2004; *Science* **348**:6232, 2015), our data do not directly address this mechanism. However, assuming strand switching is indeed the mechanism that results in re-zipping, our data suggest that the process of EcUvrD-catalyzed re-zipping can easily displace EcSSB since the rate of re-zipping is similar in the presence or absence of EcSSB.

8) See ii) above. The individual fluorescent and kymograph images (Fig. 3d,e) visualize Cy3-EcSSB binding to tracts of DNA that have been unwound by EcMutL-EcUvrD. At low concentrations of EcSSB (≤ 10 nM, left) the Cy3-EcSSB binding tracts are not saturated. The visual result is long tracts of fluorescent Cy3-EcSSB that because the smTIRF system is diffraction limited cannot resolve regions where the displaced strand contains gaps that are not bound by Cy3-EcSSB. Similar long tracts of unwound DNA are observed in the absence of EcSSB (Fig. 1) and are consistent with the conclusion that unwound ssDNA which is not bound or contains gaps that are not bound by EcSSB, allows additional (multiple) EcUvrD helicase molecules to bind and track along the displaced ssDNA strand ultimately inhibiting re-zipping. In contrast, at saturating concentrations (≥ 200 nM, right) the DNA that is unwound by EcMutL-EcUvrD is fully coated by EcSSB, which appears to block additional binding of EcUvrD ultimately resulting in more frequent re-zipping. The visual result at saturating concentrations is very short tracts of fluorescent Cy3-EcSSB that appear (unwinding) and disappear (re-zipping) over time. We have altered the text to better explain this experiment and its observations.

9) Referee #1 makes an important point that we have previously encountered with the human single stranded binding protein RPA and EXO1 during MMR excision. In this case, the ssDNA that is bound by RPA following EXO1 digestion has virtually the same length as duplex DNA. This means that even though ssDNA is formed, we could not detect it by smFS when it is bound by RPA. For the present studies, we have previously shown that ssDNA bound by EcSSB is easily distinguished from duplex DNA (*Proc Nat Acad Sci USA* **113**:3281, 2016). However, if ssDNA-EcSSB protein binding characteristics can be altered by exonucleases, it is formally

possible that we would be unable to detect excision products by smFS. We now address this possibility experimentally. First, there has been no evidence over the decades that any of the *E.coli* ssDNA exonucleases can remain bound to ssDNA alone. Thus, in order to have an obscuring property in smFS, the exonuclease(s) would need to bind EcSSB and alter its binding characteristics such that the formation of ssDNA would be disguised. Additional data now included in Fig. 5c,d shows that we can *directly* detect non-excision and excision events in real-time by monitoring Cy3-EcSSB bound to unwound ssDNA formed by EcMutL-EcUvrD unwinding events by smTIRF (Fig. 5c). If the *E.coli* ssDNA exonuclease(s) altered the EcSSB binding characteristics to obscure excision during smFS, we should observe an increase in Cy3-EcSSB that remains bound to the mismatched DNA over the entire 25 min observation period by smTIRF (indicating excision). In fact, we observe a nearly identical frequency of molecules that contain stably bound Cy3-EcSSB in the presence or absence of the *E.coli* ssDNA exonucleases. These results further suggest that extensive excision events are rare and not obscured in the smFS system.

10) Our studies are consistent with the conclusion that EcUvrD can only be captured when it forms an incipient unwinding structure at a strand scission. Thus, we observe no interactions between EcMutL and EcUvrD in the absence of an ssDNA break. This means that there is unlikely to be an interaction between EcMutL and EcUvrD prior to the formation of EcMutS-EcMutL/EcMutH strand scission at a hemimethylated GATC site. Our previous studies demonstrated that the lifetime of the EcMutL sliding clamp (14 min) was ~5-fold longer than the EcMutS sliding clamp (~3 min) and the EcMutL-EcMutH complex (~3 min; *Nature* 539:583, 2016). It seems likely that once a strand scission is introduced by EcMutS-EcMutL/EcMutH, the EcUvrD may begin incipient unwinding events (*Science* 348:6232, 2015) that will ultimately be captured by EcMutL sliding clamps in between EcMutH association-dissociation events. Moreover, the overwhelming concentration of EcUvrD should significantly drive an EcMutL interaction, but only after the EcMutS-EcMutL/EcMutH strand break is introduced and EcMutL is free of EcMutH. We have revised the text to better explain this process.

11) The evolutionary problem of MMR is that all other organisms outside the small family of γ -proteobacteria that includes *E.coli*, utilize an endogenous MutL endonuclease (MutL-endo) as well as interaction with the replication processivity factor β -Clamp (or PCNA in eukaryotes). The MutL-endo/ β -Clamp combination appears to be a functional equivalent of MutH and UvrD (*DNA Repair* 38:32, 2016). To date, no organisms containing the three MutL-endo, MutH and UvrD components have been identified. This seems to suggest that either an “intermediate” organism ultimately disappeared from biology, or the loss of the MutL-endo simultaneously occurred with mutation(s) that promoted MutL interaction with MutH and UvrD. While exclusion of individual interactions during such evolutionary development is indeed possible, if EcMutH and EcUvrD utilize similar peptide interaction motifs/mechanics, then a single (or few) mutational changes could in theory simultaneously conscript both protein-binding activities for MMR. We view this to be just one hypothesis that may be tested by future experiments and have revised the text accordingly.

12) Efficiency in biology is very hard to quantify and/or predict. Our studies have demonstrated numerous stochastic diffusion and interaction steps by MMR components that also appear to include randomly removing DNA segments between adjacent GATC strand scissions, some of which do not contain the mismatch. From an anthropomorphic point of view these numerous stochastic processes may appear to be inefficient. However, from a survival and genome maintenance point of view, MMR must just get it done. Growing evidence is suggesting that many biochemical processes rely on stochastic mechanics (see for example: *J Mol Biol. special issue* Vol. 430, 2018). It is possible that these types of mechanisms may be the only way to

complete biochemical reactions in a molecular environment where rampant thermal motion is both destructive and necessary – making them inherently the most efficient mechanisms available to biology.

Referee #2

We agree entirely with Referee #2 in his/her suggestion that “strong claims require strong evidence”. The revised manuscript includes significant additional data that provides further convincing evidence for our conclusions.

1) Referee #2 is only partially correct in his/her assertion that there is “no *in vivo* evidence” that MMR employs extremely stable MutS and MutL sliding clamps. Biteen and colleagues have examined *B.subtilis* (Bs) BsMutS *in vivo* utilizing innovative single molecule cellular imaging techniques (*Proc Nat Acad Sci USA* **112**:E6898, 2015). These studies found that BsMutS exhibited multiple distinct diffusion forms within the cell, two of which were most consistent with mismatch recognition followed by a transition to a sliding clamp. Similar studies are underway for the MutL homologs.

In addition, we note that while Hombauer et al. (*Cell* **147**:1040, 2011) observed few foci where Msh2-Msh6 and Mlh1-Pms1 co-localized, the formation of Mlh1-Pms1 foci was absolutely dependent on functional ATP-binding by Msh2-Msh6 that is essential for sliding clamp formation. Our previous studies with the bacterial MutS and MutL homologs suggest that the lifetime of the MutL sliding clamp was significantly longer than the MutS sliding clamp (see above; *Nature* **539**:583, 2016). Preliminary single molecule-imaging work with the human homologs suggests they exhibit similar properties. Together, these observations appear consistent with a hypothesis that suggests the extended lifetime of Mlh1-Pms1 on the DNA is responsible for the distribution and timing of MMR component foci observed by Hombauer et al. Nevertheless, we have been very careful not to expand our observations to a “paradigm for MMR in general” in the absence of additional non-bacterial experiments.

2) We designed the DNA substrate utilized in these single molecule-imaging studies to be appropriate for the diffraction-limited environment of the smTIRF system. This means that without super-resolution manipulation techniques, fluorescent images should be at least ~300 nm (~1000 bp) apart to unambiguously resolve them. We chose 4.2 kb to insure sufficient time and space to resolve the motions of multiple fluorophores between the strand break and mismatch in the smTIRF system. Perhaps more importantly, there are two regions of the *E.coli* genome that contain GATC sites separated by >4 kb, suggesting the system as we designed it has intrinsic physiological relevance.

We examined DNA molecules containing a well-defined strand break on both the 3' and 5' side of the mismatch to reduce potential artifacts and to recapitulate the historical observation that MMR is bidirectional. Finally, we now include data showing the repair of a DNA substrate containing strand breaks flanking the mismatch, which Referee #2 rightly suggests is more like the physiological condition (Fig. 7).

3) We addressed the possibility of inadvertent DNA substrate nicks in 1) and 2) responses to Referee #1. We note that if inadvertent ssDNA breaks were significant, then mapping an initial interaction point between EcMutL and EcUvrD would have been virtually impossible (Suppl. Fig. 2c). We acknowledge that these random strand breaks exist. But they must be substantially less frequent than the site-specific strand scission that we specifically introduced into the DNA substrate.

4) We now include the control duplex DNA data for the smFS system that shows no excision compared to the mismatch-containing DNA substrate (Fig. 3a). In addition we include the dual-nick smFS data requested by both Referee #1 and #2 (Fig. 7).

5) We believe that the combined smTIRF and smFS single molecule imaging systems far exceed the resolution and kinetic capability (<100 msec) of decades old bulk biochemical systems (Modrich assay). Moreover, these single molecule studies were performed with a 5' nick, a 3' nick and nicks flanking the mismatch as requested. Finally, while examining numerous functional mutations of the MMR components may be of interest, these studies are well beyond the scope of this manuscript and are unlikely to add to the conclusions.

6) The ability of EcUvrD to switch strands during catalytic unwinding has been described in detail (*Proc Nat Acad Sci USA*. 101:6439, 2004). This allows EcUvrD to maintain 3'→5' unwinding polarity, yet move from a 5' strand break back to the mismatch. Similar strand switching by EcUvrD is evident in the smTIRF and smFS systems described here as unwinding and re-zipping events on a single DNA molecule (see Fig. 2d and Fig. 3b).

7) The smFS studies have now been performed with a 5' nick, a 3' nick and nicks flanking the mismatch.

8) The near identical catalytic rate of EcUvrD unwinding and re-zipping strongly suggests an active enzymatic mechanism that displaces EcSSB. Thus, there is no reason to expect EcUvrD will be "hampered" during MMR strand displacement. A similar ability to actively disassemble protein and DNA complexes has been described for other RecQ (EcUvrD) homologs including RAD54 (see: *Genes Dev*. 20:2479, 2006).

9) There is no evidence that EcMutL forms a stable complex with any of the *E.coli* ssDNA exonucleases. We believe that Referee #2 is referring to the eukaryotic MLH1-PMS1(PMS2) and EXO1 that do indeed appear to interact.

10) We are confused by the suggestion of a MutL-exonuclease since the *E.coli* EcMutL does not contain an intrinsic exonuclease (or endonuclease) nor does it interact with any of the ssDNA exonucleases. Nevertheless, we regard it likely that the transient release of the displaced ssDNA end by EcUvrD (or the tethered EcMutL-EcUvrD complex) allows the digestion of a few (<50) nucleotides that will ultimately inhibit ligation until the MMR process is complete.

Referee #3

We have attempted to revise the manuscript for a wider audience and to more clearly explain how conclusions were drawn. In addition we have revised the Discussion according to the suggestions of Referee #3. We have included additional data suggested by Referee #3 that visualize ssDNA regions with Cy3-EcSSB to more directly analyzed unwinding and re-zipping events by smTIRF (Fig. 3 and Fig. 5). Moreover, we have been careful to included data on the number of molecules analyzed in all figures and supplementary figures where single molecule data is presented.

Minor Comments

1) We have removed "terrestrial".

2) We have noted in several places within the text that MMR mechanisms are largely based on reconstitution studies and that there have been numerous MMR mechanisms proposed.

- 3) Most of the text referring to MutS and MutL homologs has been removed, with a focus on *E.coli* MMR.
- 4) We have re-crafted the sentence as well as referencing to make it clear that we are referring to labeling conditions described in Ref. 31-34.
- 5) We have added additional text describing how a DSB may be formed from long excision tracts.
- 6) We have added text suggesting the reader compare the representative kymographs in Fig. 1c with the representative kymographs in Fig. 3a.
- 7) We have substituted “minimizes” for “regulates” to reduce the possibility that the reader might interpret the role of EcSSB to be an “active process”.
- 8) “Historical” has been deleted.
- 9) We have deleted “hypothetically”.
- 10) Sentence deleted.
- 11) We have revised this sentence to more accurately reflect the literature.
- 12) The space between Nb. and BbvCI was a typo. The correct nomenclature from New England Biolabs is Nb.BbvCI.
- 13) Supplementary Table 2 has been changed to “Mutation Rate (10e-8)”.

REVIEWERS' COMMENTS:

Reviewer #1 (Remarks to the Author):

The authors have responded to my comments and clarified the issues I raised. They have also prepared new DNA constructs and performed additional measurements and analysis to answer some of my questions in the first report. They have also added more detail to certain parts of the method section to increase clarity for non-experts and have added comparison with mismatch repair in other organisms in the Discussion section. They have also discussed possible reasons for conflicting results between their measurements and bulk assays and also other single molecule assays that use different DNA constructs. Even though there might still be some open questions due limitations of the techniques, I believe the authors have made a respectable effort to respond to the cumulative of reviewer reports. The mismatch repair model proposed by the authors and the compelling evidence they provide are significant and would be of broad scientific interest.

Reviewer #2 (Remarks to the Author):

Fishel Nat Commun. 2019 R1

The revised manuscript is significantly improved and the proposed model now is more convincingly supported. Nevertheless, I still feel that the single molecule approach is reductionist and should be complemented by other (biochemical and in vivo) approaches to demonstrate its physiological relevance. I have a few remaining gripes with respect to both the rebuttal letter and the revised text.

1. The response to Reviewer 1 states that "the cellular concentrations of EcSSB and EcUvrD are ~5 μ M and ~3 μ M respectively: This is not substantiated by any data or reference, and grossly incorrect.

Rather, the cellular EcSSB concentration is 0.5-1 microM (PMID:6272273) which equals to 800 tetramers/cell (Kornberg, DNA replication handbook) whereas that of EcUvrD is 3000 molecules=1500 dimers per cell (PMID:6319240).

2. The Rebuttal and revised text provide conflicting and internally inconsistent numbers of the frequency of unwanted ssDNA and dsDNA (after helicase-mediated unwinding) breaks. Thus, the original text claims that unwinding nearly always caused a dsDNA break. In the Revision this is variably changed to sometimes, often, producing a small background of unwinding events and (in the Rebuttal) the term sporadic is used. This is confusing and, as the number of inadvertent nicks may affect the interpretation of the results, should be substantiated with a more precise quantification (or estimation) of the number of such nicks and resulting dsDNA breaks, e.g. from the number of excision tracts initiation at the wrong position, or from the frequency of dsDNA breaks caused by helicase activity opposite such nicks.

3. The heading (in the Results) that the number of GATC sites implicates helicase unwinding in MMR is an overstatement. At best it is consistent with helicase unwinding.

4. In contrast to the proposed requirement of GATC methylation (by Dam) and nicking of hemimethylated Dam sites by MutH, Dam and mutH strains show only a partial MMR defect and, moreover, the mild mutator phenotype of a dam mutant is independent of MutS (e.g. PMID: 28107644). Thus, not all MMR is Dam or MutH-dependent. Therefore, in addition to GATC sites here may be other sources of nicks that can direct MMR to the newly-synthesized DNA strand: the BER-mediated incision of inadvertently introduced ribonucleotides. This has been shown in mammals. And, at the lagging strand, the nicks may be provided by the Okazaki fragments. These possible alternative and relevant sources of nicks might be mentioned in the text.

Reviewer #3 (Remarks to the Author):

In this revision, the authors have done an effective job addressing my prior comments. Overall, I consider this to be a well-done study that advances the field and sheds new light on the mechanism of *E. coli* MMR, which is much less well understood than most readers might think. As such, it's a significant contribution. I have a few minor comments that if addressed through a small amount of rewriting would help the paper.

P3, L 69, 70. Poor binding is not explained well. Is this true for any possible methylated, hemi-methylated, non-methylated state? If so, this doesn't seem relevant.

P7, L173 and throughout. What do the authors mean by "ensemble"? In a particular constant combination of proteins? If so, please define.

P12, L286, Fig. 6b. Including a plot of the Exo-independent data in this panel would help make the point.

P18, L399. Citing Goellner *Mol Cell* 2014 would be appropriate here.

I generally think each legend should list the concentrations of the proteins used in each basic reaction would be helpful.

RESPONSE TO SPECIFIC REFEREE COMMENTS:

Reviewer #1

No remaining concerns

Referee #2

To address the referee's comments regarding the "reductionist" nature of our single molecule imaging studies, we have inserted a sentence in the first paragraph of the Results comparing the original MMR reconstitution system to the one used here. In addition, we have included several sentences in the Discussion considering the possibility that other *E.coli* exonucleases might contribute to strand-specific excision during MMR as well as included data and cited numerous genetic studies that support the physiological relevance of our conclusions. We hope that these additions will further satisfy Referee #2 "reductionist" concerns?

1.) The calculations of Referee #2 for *tetramers* of EcSSB and *dimers* EcUvrD are correct and exactly the numbers that we have used in our manuscript. However, we report *monomer* concentrations since: 1) that is how biochemical studies generally refer to protein concentrations, and 2) with respect to EcUvrD there is some controversy as to whether it exists as a monomer or a dimer during catalysis.

2.) It is exceedingly difficult to quantify the frequency of random strand breaks within our 18.4 kb mismatched DNA substrate. Since we were able to localize a relatively consistent start site for EcMutL-EcUvrD unwinding events to very near the defined strand scission in this diffraction-limited system, it seems that these random ssDNA breaks are relatively infrequent. We have noted that the frequency of DSBs is clearly related to at extent of helicase unwinding. Thus, for very *long* unwinding tracts (>4-5 kb) we frequently observed a DSB. This observation suggests that either random ssDNA breaks exist or that when ssDNA tracts become extensive mechanical sheering forces may trigger a DSB. Nevertheless, we have tried to provide a *qualitative* analysis of the frequency of random ssDNA scissions (DSBs) by using qualifiers like "frequent", "occasionally" or "rarely". We hope that this estimation will be acceptable.

3.) All headings in the Results section were shortened to conform to the Editorial Requests of *Nature Communications*. That said, we believe our studies provide compelling evidence for an EcUvrD helicase strand displacement mechanism for *E.coli* strand-specific excision.

4.) In response to the original critiques of Referee #3 we had inserted text suggesting that MMR might be linked to the replication machinery enhancing the kinetics of MMR. We have now included additional text suggesting that strand breaks associated with replication could also enhance MMR and be utilized by an EcMutL-EcUvrD strand displacement mechanism as suggested by Referee #2.

Referee #3

1.) We have changed the text regarding MutH to better explain its relative binding to methylated,

hemimethylated and unmethylated GATC sites.

2.) We have largely deleted the word “ensemble” and now only use it to describe the complete MMR reaction, with explanations therein.

3.) We subtracted the exonuclease-independent from the exonuclease-dependent data to arrive at the numbers shown in Fig. 6 (see Methods). Thus, an additional plot really makes no sense.

4.) Goeliner et al., (2014) has now been cited.

5.) We have now included protein concentrations in all of the Figure Legends for clarity.